# Cellular characterization of the mouse collecting lymphatic vessels reveals that lymphatic muscle cells are the innate pacemaker cells

**Scott D Zawieja[1]\***, **Grace A Pea[1]**, **Sarah E Broyhill[1]**, **Advaya Patro[1]**, **Karen H Bromert[1]**, **Charles E Norton[1]**, **Hae Jin Kim[1]**, **Sathesh Kumar Sivasankaran[2]**, **Min Li[1]**, **Jorge A Castorena-Gonzalez[3]**, **Bernard T Drumm[4]**, **Michael J Davis[1]\***

[1]Department of Medical Pharmacology & Physiology, University of Missouri, Columbia, United States; [2]Bioinformatics and Analytics Core, Division of Research, Innovation and Impact, University of Missouri, Columbia, United States; [3]Department of Pharmacology, Tulane University, New Orleans, United States; [4]Smooth Muscle Research Centre, Dundalk Institute of Technology, Dundalk, Ireland

**\*For correspondence:**
zawiejas@umsystem.edu (SDZ);
DavisMJ@health.missouri.edu (MJD)

**Competing interest:** The authors declare that no competing interests exist.

## eLife Assessment

This manuscript aims to identify the pacemaker cells in the lymphatic collecting vessels - the cells that initiate the autonomous action potentials and contractions needed to drive lymphatic pumping. Through the exemplary use of existing approaches (genetic deletions and cytosolic calcium detection in multiple cell types), the authors **convincingly** determine that lymphatic muscle cells are the origin of the action potential that triggers lymphatic contraction. The inclusion of scRNAseq and membrane potential data enhances a tremendous study. This **fundamental** discovery establishes a new standard for the field of lymphatic physiology.

**Abstract** Collecting lymphatic vessels (cLVs) exhibit spontaneous contractions with a pressure-dependent frequency, but the identity of the lymphatic pacemaker cell is still debated. Here, we combined immunofluorescence and scRNAseq analyses with electrophysiological methods to examine the cellular constituents of the mouse cLV wall and assess whether any cell type exhibited morphological and functional processes characteristic of pacemaker cells. We employed inducible Cre mouse models to target-specific cell populations including CkitCreER[T2] to target interstitial cells of Cajal-like cells, PdgfrβCreER[T2] to target pericyte-like cells; PdgfrαCreER[TM] to target CD34[+] adventitial cells; and Myh11CreER[T2] to target lymphatic muscle cells (LMCs) directly. These inducible Cre lines were crossed to the fluorescent reporter $Rosa26^{mTmG}$, the genetically encoded Ca[2+] sensor GCaMP6f, and the light-activated cation channel rhodopsin2 (ChR2). Only LMCs consistently, but heterogeneously, displayed spontaneous Ca[2+] events during the diastolic period of the contraction cycle, and whose frequency was modulated in a pressure-dependent manner. Further, optogenetic depolarization with ChR2 induced propagated contractions only in LMCs. Membrane potential recordings in LMCs demonstrated that the rate of diastolic depolarization significantly correlated with contraction frequency. These findings support the conclusion that LMCs, or a subset of LMCs, are responsible for mouse cLV pacemaking.

## Introduction

The spontaneous contractions of collecting lymphatic vessels (cLVs) are an integral component to fluid and macromolecule homeostasis as they provide the force to transport fluid from the interstitial spaces back to the blood circulation (*Scallan et al., 2016*). In humans, spontaneous contractile activity is estimated to account for 2/3 of lymph transport (*Engeset et al., 1977*), and this function is significantly compromised in patients suffering from lymphedema, whose cLVs typically display weak and irregular or entirely absent contractile activity (*Olszewski, 2002*). Ex vivo studies, in which the intraluminal pressure can be precisely controlled, have refined our understanding of the pressure-dependent regulation of contraction frequency (*Benoit et al., 1989*; *Gashev et al., 2004*), with some mouse cLVs displaying a 10-fold increase in contraction frequency over a 10 cmH$_2$O pressure gradient (*Scallan and Davis, 2013*; *Zawieja et al., 2018a*). The observation that cLVs, often cannulated at various lengths for ex vivo preparations, retain a consistently tunable contraction frequency points to the presence of (a) pacemaker cell(s) innate to the structure of the cLV wall and with a seemingly ubiquitous presence along the length of the vessel (*Zawieja et al., 1993*; *Castorena-Gonzalez et al., 2018b*). Furthermore, isolated cLVs typically display single pacemaker initiation sites unless damaged or electrically uncoupled by pharmacological inhibition of gap junctions or genetic deletion of *Gjc1* (also known as *Connexin 45, Cx45*) (*Behringer et al., 2017*; *Castorena-Gonzalez et al., 2018b*; *Castorena-Gonzalez et al., 2020*). In sum, this suggests the pacemaker cell(s) is(are) likely both ubiquitous and continuous, to allow for electrical conduction via gap junctions, along the length of the cLV and prevent colliding contractile waves which would impair lymph transport.

Investigations into the cLV pacemaker identity have focused largely on cells termed interstitial cells of Cajal-like cells (ICLCs; or telocytes) (*McCloskey et al., 2002*; *Briggs Boedtkjer et al., 2013*), as they display some morphological and cell marker expression profiles similar to the interstitial cells of Cajal (ICC), which are bona fide pacemakers in the gastrointestinal (GI) tract. ICC are classically identified by either methylene blue staining and expression of CKIT, and coordinate GI smooth muscle contraction (*Maeda et al., 1992*; *Ward et al., 1994*; *Ordög et al., 1999*). ICC also expresses the canonical Ca$^{2+}$-activated Cl$^-$ channel *Anoctamin 1* (*Ano1*) (*Gomez-Pinilla et al., 2009*), which is required for pacemaker activity (*Hwang et al., 2009*; *Zhu et al., 2009*; *Singh et al., 2014*). Previous reports in sheep mesenteric lymphatic vessels identified a population of CKIT$^+$ and VIMENTIN$^+$ ICLC in the vessel wall between the endothelial and lymphatic muscle cell (LMC) layer and running along the axis of the vessel (*McCloskey et al., 2002*). Investigations in the human thoracic duct also identified a significant population of ICLCs in close proximity to the LMCs evident by methylene blue staining, immunostaining for CD34, VIMENTIN, and CKIT, as well as the gold standard of electron microscopy (*Briggs Boedtkjer et al., 2013*). However, neither study could determine if these cells had functional electrical communication with the LMCs or demonstrate either a membrane electrical clock or internal Ca$^{2+}$ clock to drive the rhythmic lymphatic vessel contractions observed ex vivo. LMCs share a functional similarity to ICC in that they also display the ANO1-mediated Ca$^{2+}$-activated Cl$^-$ current (*van Helden, 1993*; *Toland et al., 2000*; *Mohanakumar et al., 2018*; *Zawieja et al., 2019*), that regulates pacemaking. Spontaneous transient depolarizations, presumably ANO1 dependent, were recorded in mesenteric cLVs from guinea pigs (*van Helden, 1993*; *von der Weid et al., 2008*), providing a mechanism for membrane potential instability to drive AP initiation. Furthermore, computational models have proposed LMC sarcoplasmic reticulum (SR) Ca$^{2+}$ release as the oscillator mechanism driving pacemaking (*Imtiaz et al., 2007*). SR Ca$^{2+}$ release has also been implicated in pericyte regulation of arterioles (*Hashitani et al., 2015*; *van Helden and Imtiaz, 2019*), in microvascular vasomotion (*Boedtkjer et al., 2008*; *Aalkjær et al., 2011*; *van Helden and Imtiaz, 2019*), and in the contraction waves of atypical muscle cells of the lower urinary tract (*Grainger et al., 2022*).

Presently, no investigations have clearly identified the cellular identities of possible pacemaker cells within the cLVs of the mouse. Mouse cLVs exhibit contractile parameters and conduction speed equivalent to those of human vessels (*Castorena-Gonzalez et al., 2018b*) and their simplified architecture, compared to larger mammals, in combination with the genetic tools developed for the mouse model, allowed us to test for a fundamental pacemaker cell in the cLV. In this study, we utilized multiple genetic mouse models, confocal imaging of fluorescent reporters, cell-specific expression of GCaMP6f for Ca$^{2+}$ imaging, and optogenetic light-activated depolarization to both visualize and test the functional aspects of putative pacemaker cells, along with membrane potential recordings in LMCs in pressure-challenged cLVs. We also performed immunostaining and single-cell RNA sequencing (scRNAseq)

of isolated cLVs to provide greater detail to the heterogenous cellular populations found within the mouse cLVs. Despite identifying a significant population of CD34⁺PDGFRα⁺ adventitial cells along the length of mouse cLVs, the results of our functional studies support a myogenic (LMC) origin of pacemaking in cLVs.

## Results

### Methylene blue staining reveals a minor population of cells in mouse cLVs

Methylene blue staining was used to identify an ICLC population in the human lymphatic thoracic duct (*Briggs Boedtkjer et al., 2013*). In our isolated and cleaned lymphatic inguinal axillary collecting vessels (IALVs), methylene blue stained a significant number of cells with variable density along the length of the IALV and heterogeneous cell morphologies (*Figure 1A–C*). A significant portion of the stained cells resembled lymphatic vessel-associated macrophages with an elongated shape, while other cells were smaller and circular (*Figure 1D–F*). Methylene blue also appeared to stain mast cells as there were large ovoid cells with intracellular granules on the adventitial surface of the vessel. In addition, methylene blue stained a minor population of cells that exhibited long and thin axon-like extensions which appeared to have a slight helical orientation, with a small central body and nucleus (*Figure 1C*). None of these cell populations were aligned with the longitudinal axis of the vessel that would permit efficient coupling or regulation across the circumferential layer of LMCs required for coordinated propagation along the length of the vessel.

### Immunofluorescence Imaging of IALVs Stained for ICLC, LEC, and LMC Markers

We next stained IALVs for the putative telocyte/ICLC markers CKIT, CD34, and the intermediate filament VIMENTIN, which have been previously utilized for ICLC identification in human and sheep lymphatic tissues (*McCloskey et al., 2002*; *Briggs Boedtkjer et al., 2013*). Additionally, an antibody to the intermediate filament DESMIN was used to label muscle cells (*McCloskey et al., 2002*). IALVs stained for cKIT (*Figure 2B*) showed robust signal in large ovoid cells with a non-segmented circular nucleus (*Figure 2A*), characteristic of mast cells that were in the outer part of the adventitia. Similarly, CKIT-stained populations of elongated cells as well as circular cells with variable densities were observed throughout the IALV wall, similar to methylene blue⁺ cell populations (*Figure 2B, J*). Staining for CD34 revealed a large population of cells that were seemingly contiguous along the length of the vessel. The CD34⁺ cells generally had multiple lobular processes and a 'oak leaf'-like appearance, typically a characteristic of fibroblasts, though some contained short, thin dendrite-like extensions (*Figure 2C, G, K*). The CD34⁺ cells were negative for DESMIN (*Figure 2H*), which primarily stained the circumferential LMCs (*Figure 2F*). Note that the largely non-circumferential cell organization in this region is typical for a lymphatic endothelial valve site (*Bridenbaugh et al., 2013a*). Furthermore, CD34⁺ and cKIT⁺ cells were separate populations (*Figure 2D, L*). A VIMENTIN antibody labeled lymphatic endothelial cells (LECs) which exhibited a horizontal cobblestone morphology in parallel with the vessel axis (*Figure 2E, I*), while also co-labeling the majority of the CD34⁺ cells (*Figure 2H*) and CKIT⁺ cells (*Figure 2L*). Videos of the half vessel z-stacks are provided (*Figure 2—video 1*; *Figure 2—video 2*; *Figure 2—video 3* for *Figure 2D, H, L*, respectively).

Of the cells stained in *Figure 2*, the CD34⁺ population was intriguing due to its high density and distribution throughout the length of the IALV, which potentially would be conducive to effective regulation of LMC excitability. In addition to CD34 and VIMENTIN, PDGFRα staining is also commonly ascribed to both telocytes (*Vannucchi et al., 2013*; *Xiao et al., 2013*; *Zhou et al., 2015*) as well as fibroblasts (*Kimura et al., 2021*; *Clayton et al., 2022*). We performed immunofluorescence imaging for PGDFRα counterstained with CD34 and markers for LMCs, LECs, and pericytes. As noted in *Figure 2*, CD34⁺ cells (*Figure 3A*) did not co-label LMCs (*Figure 3D*), which were smooth muscle actin⁺ (ACTA2, *Figure 3B*) and CALPONIN⁺ (*Figure 3C*). However, nearly all CD34⁺ (*Figure 3E*) cells were also PDGFRα⁺ (*Figure 3F, H*). Occasionally, some overlap of PDGFRα and ACTA2 signal was noted (*Figure 3G, H*). LECs staining with PECAM1 (*Figure 3I*) revealed the expected rectangular elongated cobblestone morphology that was distinct from the PDGFRα⁺ cells (*Figure 3J, L*). Staining for CALPONIN also specifically labeled LMCs (*Figure 3K*) but not PDGFRα⁺ cells (*Figure 3L*). Lastly,

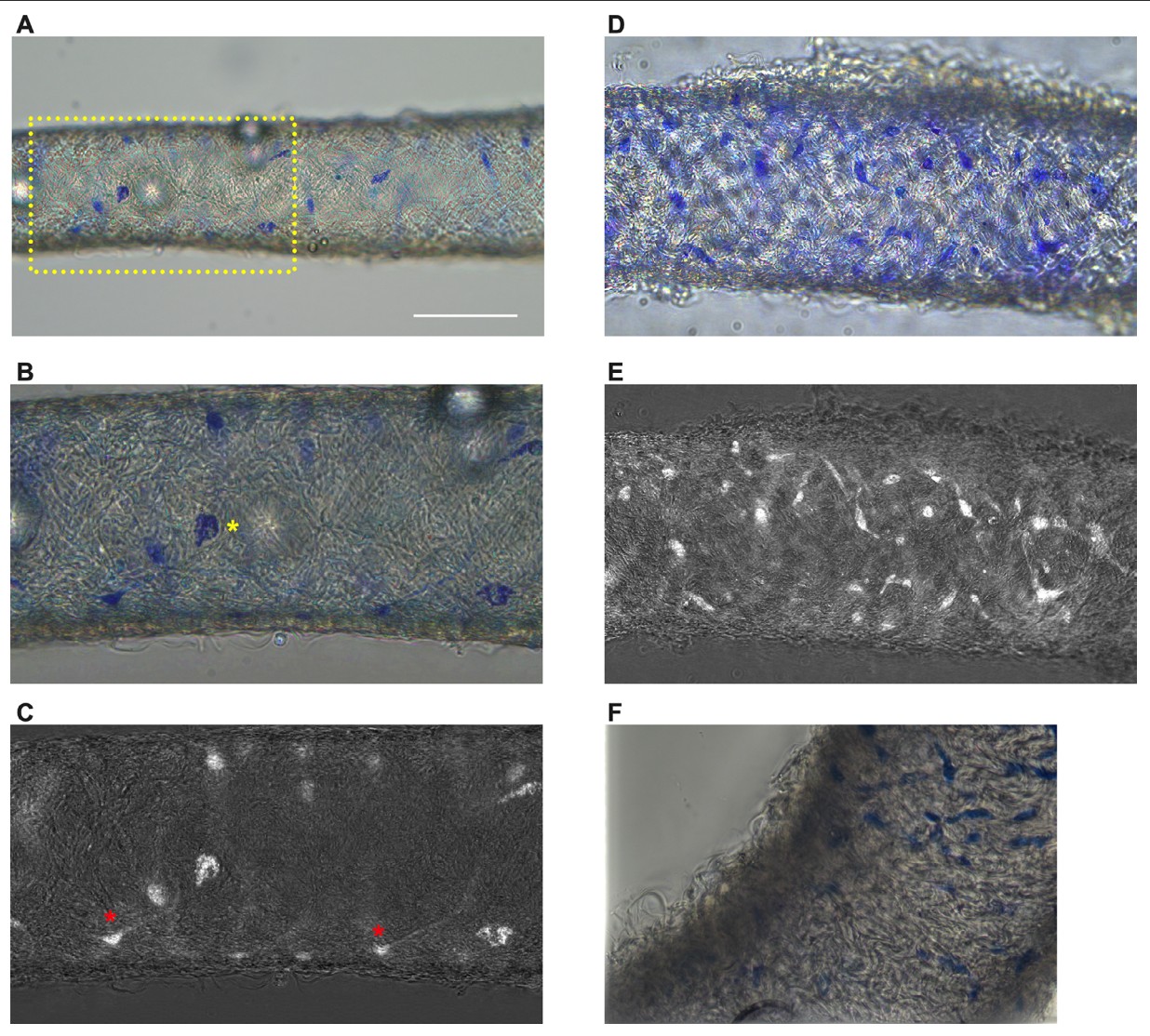

**Figure 1.** Methylene blue staining of isolated mouse IALVs. Representative image of an isolated and cleaned IALV after methylene blue staining which revealed cells of various morphology (**A**). (**B**) is the zoomed-in image of the yellow dotted box in A which contained large ovoid cells with granular staining (**B**, yellow asterisks). Fine cellular extensions (red asterisks) stained by methylene blue in some cells were visualized with color channel separation and division (**C**). (**D, E**) Similar to B and C, but in a separate vessel which stained with a higher density of methylene blue stained cells, some of which had limited cellular processes. (**F**) Focal reconstruction from imaging a methylene blue stained IALV using an upright microscope and immersion objective. Methylene blue staining was performed in IALVs isolated from five mice (*n* = 5).

we stained for PDGFRα, CD34, and the commonly used pericyte marker PDGFRβ (*Figure 3M–P*). As above, CD34 and PDGFRα were highly colocalized (*Figure 3Q, R, T*), and many of the CD34+ and PDGFRα+ cells were also PDGFRβ+ (*Figure 3P*). PDGFRβ also stained some circumferential LMCs (*Figure 3Q*). During the imaging of mouse IALVs for these markers, we also observed that the lymphatic secondary endothelial valves were populated by elongated cells that stretched the length of the valve leaflet and were positive for CD34, PDGFRα, and PDGFRβ, with varying intensities. These cells could be observed in most, if not all, the valves we assessed and found within both leaflets of the valve (*Figure 3R, S*). These cells had long, thin extensions that were branched, along with apparent dendrite-like extensions with a morphology that closely resembled those described for pericytes or telocytes (*Popescu and Faussone-Pellegrini, 2010*). PDGFRα+ or CD34+ cells with this morphology were only observed in the valve leaflets, and thus seemed insufficient to regulate pace-making as normal contractions are observed in isolated cLVs without secondary valves (*van Helden,*

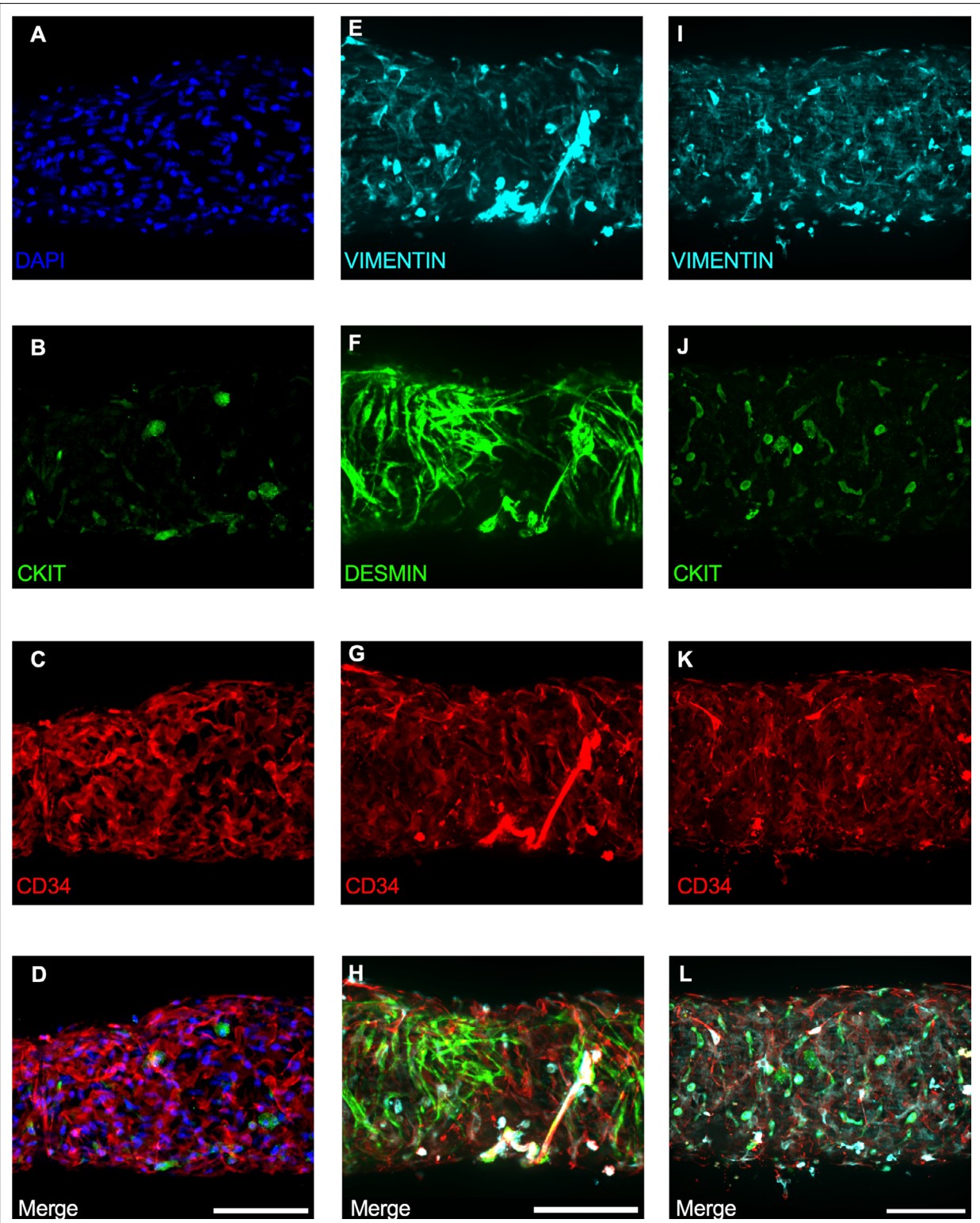

**Figure 2.** Staining mouse IALVs for ICLC markers. Representative immunofluorescent max projections of half vessel confocal image stacks imaged from mouse IALVs stained for ICLC markers. DAPI (**A**), cKIT (**B**), and CD34 (**C**) and their merged image (**D**). Representative max projections of the intermediate filament VIMENTIN (**E**), the intermediate filament DESMIN (**F**), CD34 (**G**), and their merged image (**H**). Representative max projection of VIMENTIN (**I**), CKIT (**J**), CD34 (**K**), and their merged image (**L**). Scale bar = 100 μm for all images.

The online version of this article includes the following video(s) for figure 2:

*Figure 2 continued on next page*

*Figure 2 continued*

**Figure 2—video 1.** Expression of ICCLC markers in mouse IALVs.
https://elifesciences.org/articles/90679/figures#fig2video1

**Figure 2—video 2.** Expression of DESMIN in lymphatic muscle cells (LMCs) and CD34 in AdvCs.
https://elifesciences.org/articles/90679/figures#fig2video2

**Figure 2—video 3.** CKIT and CD34 label separate cell populations in mouse IALVs.
https://elifesciences.org/articles/90679/figures#fig2video3

*1993*; *Gashev et al., 2002*). Representative z-stacks demonstrating these valve-located 'telocyte' shaped cells (*Figure 3R, S*) are provided in *Figure 3—videos 1 and 2*.

We next determined the degree of colocalization between the CD34 and PDGFRα signal given the significant overlap in their staining profile. Colocalization analysis of PDGFRα (*Figure 3—figure supplement 1A*) and CD34 (*Figure 3—figure supplement 1B*) and their colocalization (*Figure 3—figure supplement 1C*) was determined with the FIJI BIOP-JACoP tool. The Pearson's coefficient was 0.83 (*Figure 3—figure supplement 1D*) and Mander's coefficient of overlap was 0.80 for the PDGFRα$^+$ signal and 0.87 for the CD34 signal (*Figure 3—figure supplement 1E*). Colocalization between MYH11 and PDGFRα was significantly lower (*Figure 3—figure supplement 1D–F*) with a Pearson's coefficient of 0.30 (*Figure 3—figure supplement 1G*), whereas the Mander's coefficient for MYH11 overlap with PDGFRα was 0.077 and 0.043 for PDGFRα signal overlap with MYH11 (*Figure 3—figure supplement 1H*). The high degree of colocalization of CD34 and PDGFRα signal informed our use of the commercially available transgenic PdgfrαCreER$^{TM}$ mouse model to target these cells. The vast majority of the PDGFRα$^+$ cells were located in the adventitial layer (*Figure 3—figure supplement 2A–D*), which varied between 1 and 3 PDGFRα$^+$ cells thick (*Figure 3—figure supplement 2E*). Under this layer, we observed only a single layer of largely circumferential LMCs stained by MYH11 (*Figure 3—figure supplement 2B*) sitting atop a single layer of PECAM1$^+$ LECs (*Figure 3—figure supplement 2A*). We also observed occasional PDGFRα$^+$ cells or their extensions located in the sub-endothelial space (*Figure 3—figure supplement 2E', E''*) positioned between the LECs and the LMCs.

## Use of inducible Cre-mediated recombination of *Rosa26$^{mTmG}$* to delineate and characterize specific IALV cell types

After confirming the presence of VIMENTIN$^+$, cKIT$^+$, and CD34$^+$ PDGFRα$^+$ positive cells within the mouse IALV, we sought to further investigate these cell populations by using constitutive and inducible Cre recombinase expressing mouse lines. IALVs from the constitutively active PdgfrαCre-*Rosa26$^{mTmG}$* and Cspg4Cre-*Rosa26$^{mTmG}$* mice had GFP fluorescence in the majority of LMCs as well as in the fibroblast-shaped cells found within the IALV wall (*Figure 4A, B*). While informative of expression of the LMC progenitor cells, neither constitutive Cre would be useful in delineating cell types. In contrast to the constitutively active PdgfrαCre, the tamoxifen-inducible PdgfrαCreER$^{TM}$ line drove significant recombination in only the fibroblast-shaped cells previously stained for CD34 and PDGFRα but not in LMCs or LECs (*Figure 4C*). PdgfrβCreER$^{T2}$, commonly used to label pericytes, drove recombination in both a minor population of the LMCs and the fibroblast-shaped cells. CkitCreER$^{T2}$, which capably drives recombination in the ICCs of the GI tract (*Baker et al., 2016*), drove recombination only in a small population of irregularly spaced, large ovoid cells on the surface of the IALV (*Figure 4E*), although recombination in one or two LECs could occasionally be detected (not shown). Finally, Myh11CreER$^{T2}$ drove recombination in nearly all LMCs, which were largely circumferentially oriented with dendrite-like, cell-cell contacts visible between them and without significant GFP fluorescence in either LECs or the fibroblast-shaped CD34$^+$ PDGFRα$^+$ cell population (*Figure 4F*). Additionally, some LMCs maintained the bipolar shape but had secondary extensions forming a 'Y' shape in which an adjacent LMC typically filled the inner void. A very minor population of recombined cells in the Myh11CreER$^{T2}$-*Rosa26$^{mTmG}$* IALVs was smaller and irregularly patterned with multiple fine axon-like projections or ruffled edges (*Figure 4F*).

To complement the morphological and cell density findings obtained with confocal microscopy, we digested IALVs from the inducible Cre-*Rosa26$^{mTmG}$* lines, and the Prox1-eGFP line as a control, into single-cell suspensions and sorted the respective GFP$^+$ populations (*Figure 4G–J*) for RT-PCR profiling (*Figure 4K*). We first focused on determining the molecular fidelity of the sorted cells based on the

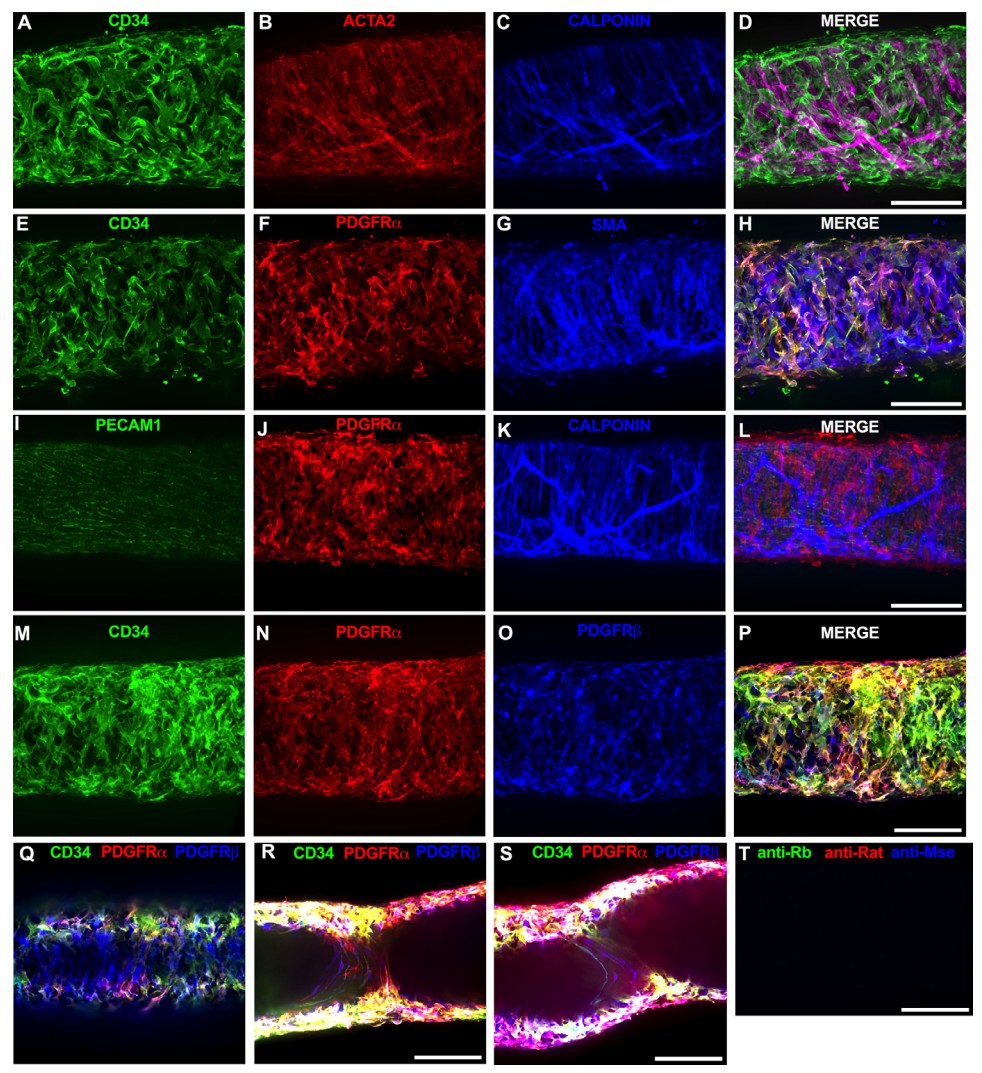

**Figure 3.** Immunofluorescence labeling of mouse IALVs with markers for ICLC, lymphatic muscle cell (LMC), lymphatic endothelial cell (LEC), and immune cell populations. We stained isolated mouse IALVs with cellular markers used to differentiate various cell types observed in collecting lymphatic vessels (cLVs). Half vessel image stacks were taken with confocal microscopy and the resulting representative max projections are shown. (**A**) CD34 stained cells and LMC staining with ACTA2 (**B**) and CALPONIN (**C**) and the corresponding merged (**D**) image. There was significant overlap in (**E**) CD34 staining along with the fibroblast marker PDGFRα (**F**) compared to LMC staining with ACTA2 (**G**) and the merged (**H**) image. The endothelial marker PECAM1 (**I**) to delineate LECs with PDGFRα staining (**J**), and the LMC marker CALPONIN (**K**) with the merged image (**L**) revealed three separate populations of cells. PDGFRβ (**O**) stained many cells that were CD34 (**M**) and PDGFRα (**N**) positive, as seen in the merge imaged (**P**), in addition to PDGFRβ signal detected in the LMC layer (**Q**). Max projections of only the luminal frames of a z-stack at lymphatic valve locations revealed PDGFRβ, CD34, and PDGFRα labeling in bipolar shaped cells with long extensions that traveled throughout the valve leaflets (**R**, **S**). Control IALV (**T**) stained only with secondary antibody. Scale bar = 100 μm for all images.

The online version of this article includes the following video and figure supplement(s) for figure 3:

**Figure supplement 1.** Colocalization of CD34 and PDGFRα.

**Figure supplement 2.** PDGFRα⁺ cells reside primarily in the mouse lymphatic collecting vessel adventitia and some in the subendothelial space.

**Figure 3—video 1.** PDGFRα, CD34, and PDGFRβ label lymphatic valve interstitial cells.
https://elifesciences.org/articles/90679/figures#fig3video1

**Figure 3—video 2.** Lymphatic valve interstitial cells have variable expression of PDGFRα, CD34, and PDGFRβ.
https://elifesciences.org/articles/90679/figures#fig3video2

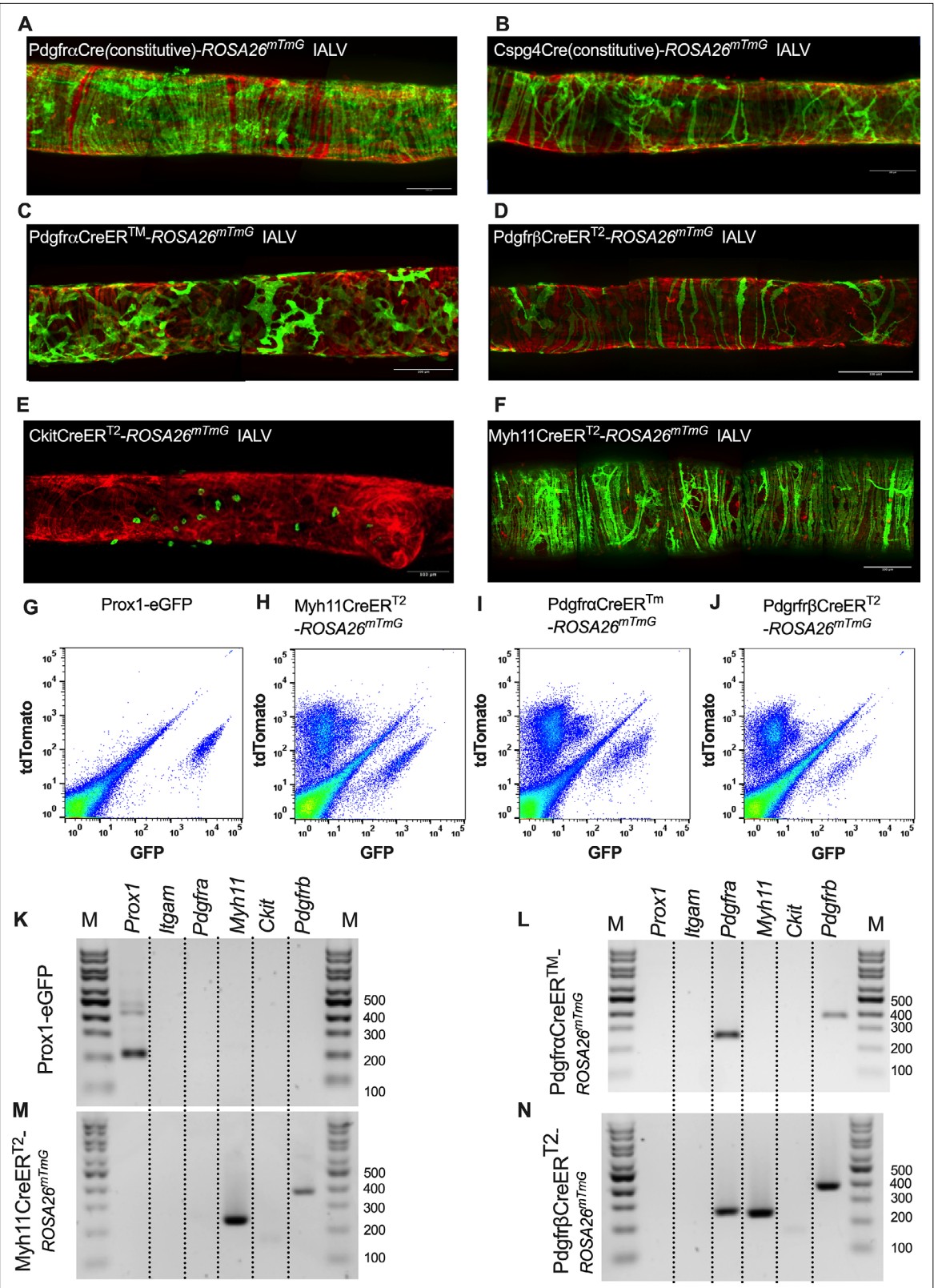

**Figure 4.** Inducible Cre-*Rosa26^{mTmG}* labeling and fidelity to target putative pacemaker cell populations. Stitched montages of serial max projections of GFP and tdTomato signal from live IALVs isolated from PdgfrαCre-*Rosa26^{mTmG}* (**A**), Cspg4Cre-*Rosa26^{mTmG}* (**B**), PdgfrαCreER^{TM}-*Rosa26^{mTmG}* (**C**), PdgfrβCreER^{T2}-*Rosa26^{mTmG}* (**D**), CkitCreER^{T2}-*Rosa26^{mTmG}* (**E**), and Myh11CreER^{T2}-*Rosa26^{mTmG}* (**F**). IALVs were digested into single cells and GFP⁺ cells were purified via FACS from Prox1-eGFP (**G**), Myh11CreER^{T2}-*Rosa26^{mTmG}* (**H**), PdgfrαCreER^{TM}-*Rosa26^{mTmG}* (**I**), and PdgfrβCreER^{T2}-*Rosa26^{mTmG}* (**J**) mice.

*Figure 4 continued on next page*

*Figure 4 continued*

Representative gels demonstrating RT-PCR products corresponding to the respective genes used in the promoter of each specific transgene employed to drive either eGFP or Cre-mediated recombination of *Rosa26^mTmG* from each GFP^+-sorted population (**K–N**) to assess fidelity. Images are representative of IALVs from at least three separate mice (*n* = 3). FACS and RT-PCR were repeated at least three times (*n* = 3 mice).

The online version of this article includes the following source data for figure 4:

**Source data 1.** This file contains the RT-PCR gel electrophoresis data for *Figure 4*.

**Source data 2.** This file contains the RT-PCR gel electrophoresis data for *Figure 4* without markup.

gene promoters used to drive each inducible Cre model to discern cellular overlap. In agreement with the confocal images, sorted GFP^+ cells from PdgfrβCreER^T2-*Rosa26^mTmG* IALVs expressed *Pdgfrb* but also *Myh11* and *Pdgfra*. In contrast, GFP-sorted cells from PdgfrαCreER^TM IALVs expressed *Pdgfra* and *Pdgfrb*, but with no detectable expression of *Myh11*. GFP^+ cells from sorted Myh11CreER^T2-*Rosa-26^mTmG* IALVs had high expression for *Myh11* as well as *Pdgfrb*, but did not express *Pdgfra*. IALVs from CkitCreER^T2-*Rosa26^mTmG* mice were not pursued for FACS due to the exceptionally sparse recombination observed along the IALV.

## Characterization of the cellular components of the mouse IALVs by scRNAseq and FACS–RT-PCR

The results from the immunofluorescence staining, *Rosa26^mTmG* reporter imaging, and FACS–RT-PCR experiments suggested that both LMCs and AdvCs can express *Pdgfrb*. To provide further clarity and detail to the cellular populations within the mouse cLV wall and potential subsets within those broad cell types, we performed scRNAseq on isolated and cleaned inguinal axillary cLVs from male and female mice. The resulting uniform manifold approximation and projection (UMAP) (*Figure 5A*) revealed a host of cell types which had three main clusters corresponding to LECs, LMCs, and AdvCs (*Figure 5A*). We assessed the expression of genes that correspond to the markers from our earlier immunofluorescence staining as well as cell identification markers commonly used within the literature to identify each cell cluster (*Figure 5B*). Cell identity was confirmed by commonly used markers (*Figure 5B*) and the top differentially expressed genes (*Figure 5—figure supplement 1A*). Feature plots for the LEC markers *Prox1* (*Figure 5C*) and *Flt4* (*Figure 5D*), LMC markers *Myh11* (*Figure 5E*) and *Cnn1* (*Figure 5F*), and the AdvCs markers *Pdgfra* (*Figure 5G*) and *Lumican* (*Figure 5H*) were quite specific for labeling their respective cell clusters. Very few *Kit* (*Figure 5I*) expressing cells were observed in accordance with our imaging results. *Pdgfrb* was observed in both LMC and AdvC clusters (*Figure 5J*) while the remaining cell clusters were of immune origin as they expressed the gene *Ptprc* encoding the hematopoietic marker CD45 (*Figure 5K*). Notably, the previous genes suggested to identify LMCs in a previous scRNASeq study (*Kenney et al., 2022*), *Dpt*, *Pi16*, and *Ackr3*, were largely absent in LMCs and instead were expressed in a minor population of AdvCs (*Figure 5—figure supplement 1B*). We provide a further sub-clustering breakdown of the LECs (*Figure 5—figure supplement 2*), LMCs (*Figure 5—figure supplement 3*), AdvCs (*Figure 5—figure supplement 4*), and a detailed expression profile of the immune cell clusters (*Figure 5—figure supplement 5*). Further assessment of the LEC subcluster included a putative lymphatic endothelial 'up valve' cell population in subcluster 8 which expressed high levels of *Prox1*, *Cldn11*, *Itga9*, *Gja4*, and *Neo1* and 'down-valve' population in cluster 6 which expressed *Clu*, *Adm*, *Gja4*, and *Lypd6* (*Figure 5—figure supplement 2C*) which mapped well to previous RNAseq datasets (*González-Loyola et al., 2021*; *Petkova et al., 2023*; *Yoon et al., 2024*; *Takeda et al., 2019*). The top differentially expressed genes in the putative up-valve population in cluster 8 included *Irx3*, *Neo1*, *Tub*, *Ano4*, and *Fxyd2*, and we noted *Cacna1e*, *Fgf14*, and *Irf1* in the down-valve cluster 6. Analysis of the LMC subclusters did not reveal any significant differences in the expression of known pacemaking-associated genes *Ano1* or *Itpr1* at our initial conditions of Log2FC of 0.5. However, we provide an overview of the typical ion channel families expressed in LMCs in *Figure 5—figure supplement 3B–I*. The AdvC cells could be further subclustered into multiple populations (*Figure 5—figure supplement 4A and C*) with little evidence of LMC gene contamination as these cells lacked *Myh11*, *Kcnma1*, and *Tagln* despite expression of *Cacna1c*, *Ano1*, and *Gjc1*. Over 75% of AdvCs expressed *Pdgfra* (*Figure 5—figure supplement 4D*) and 65% of the total AdvCs co-expressed both *Pdgfra* and *Cd34*. Our immunofluorescent colocalization of PDGFRα and CD34 staining was also supported as 72% of AdvCs expressing either *Cd34*

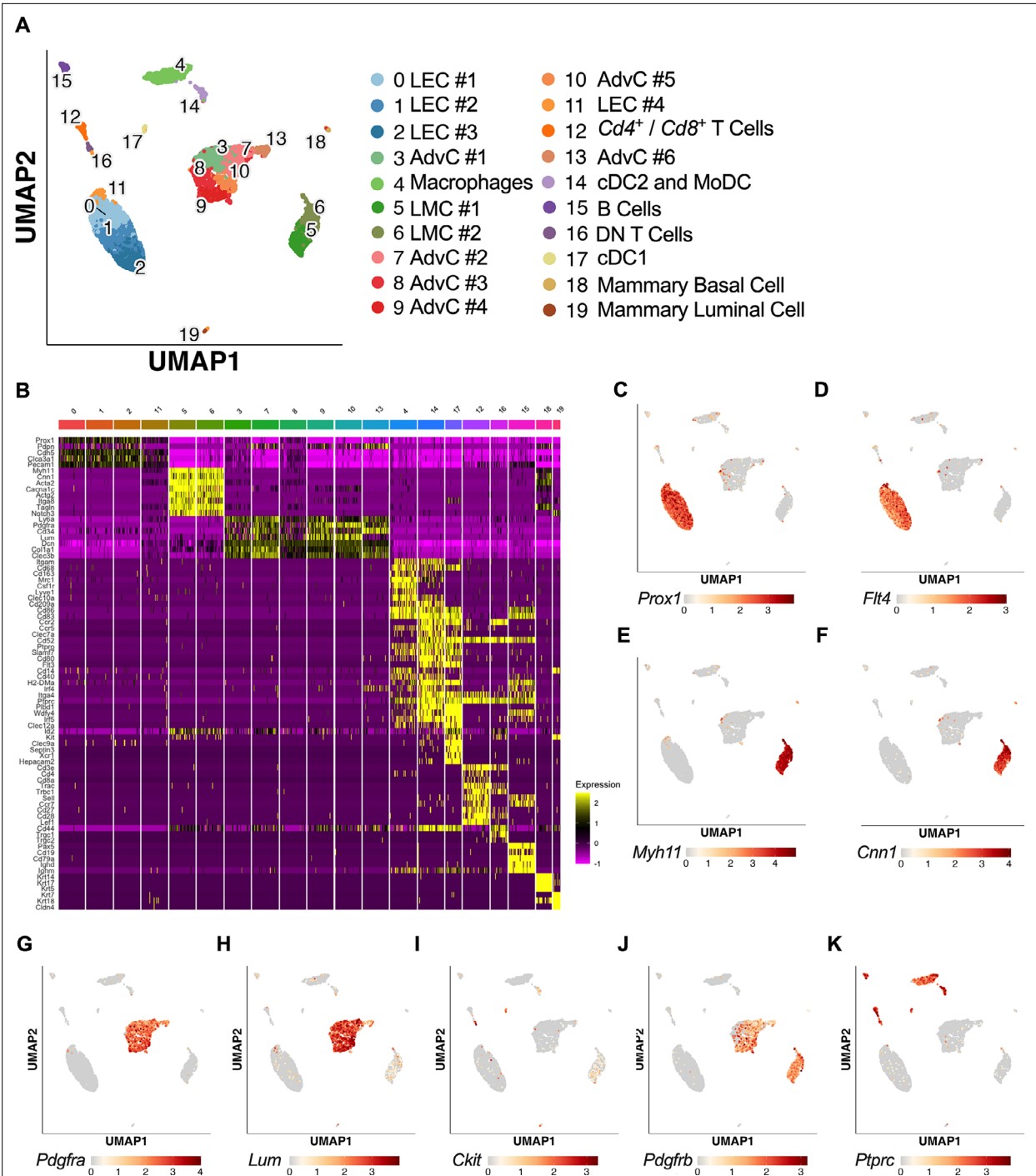

**Figure 5.** scRNAseq analysis of mouse IALVs from *Rosa26^mTmG* mice. IALVs were cleaned and isolated from 10 *Rosa26^mTmG* mice and digested into a single-cell suspension for scRNAseq analysis with the 10X platform. (**A**) Uniform manifold approximation and projection (UMAP) of the various cell populations that compromise the mouse IALV, though some mammary epithelia contamination was present (populations 18 and 19). (**B**) Heatmap of commonly used genes for cell identification for each of the cell clusters. Feature plots to assess cell cluster expression of the genes shown in *Figure 4* inlcuding the lymphatic endothelial cell (LEC) markers *Prox1* (**C**) and *Flt4* (**D**), lymphatic muscle cell (LMC) markers *Myh11* (**E**) and *Calponin 1* (**F**, *Cnn1*), fibroblast markers *Pdgfra* (**G**) and *Lum* (**H**, *Lumican*), ICC marker *Kit* (**I**), the pericyte and smooth muscle precursor marker (*Pdgfrb*) (**J**), and the hematopoietic marker *Ptprc* (**K**).

The online version of this article includes the following figure supplement(s) for figure 5:

**Figure supplement 1.** scRNAseq analysis of the mouse IALV cell populations.

*Figure 5 continued on next page*

**Figure supplement 2.** Subclusters of IALV lymphatic endothelial cells (LECs) revealed by scRNAseq.

**Figure supplement 3.** Subclusters of IALV lymphatic muscle cells (LMCs) revealed by scRNAseq.

**Figure supplement 4.** Subclusters of IALV AdvCs revealed by scRNAseq.

**Figure supplement 5.** Immune cell populations associated with the mouse IALV.

or *Pdgfra* co-expressed both genes (***Figure 5—figure supplement 5D***). The vast majority of AdvCs expressing *Pdgfrb* (***Figure 5—figure supplement 4E***) or *Cspg4* (***Figure 5—figure supplement 4F***) also expressed *Pdgfra*. Expression of *Ano1*, *Gjc1*, and *Cacna1c* was also observed in some of the AdvCs, and most of those cells also co-expressed *Pdgfra,* supporting further use of the PdgfrαCreER™ line (***Figure 5—figure supplement 4G–I***).

While scRNAseq highlighted the depth of heterogeneity of the cellular composition of the mouse cLV, we wanted to validate the actual recombined cell populations from the inducible Cre-*Rosa26^{mTmG}* lines. We profiled each inducible Cre-driven recombination of *Rosa26^{mTmG}* via FACS-purified cells and RT-PCR for common markers for endothelial cells, muscle cells, and pericytes. *Nos3* (eNOS) expression was observed only in the Prox1-eGFP sorted cells, and LECs also expressed *Vim*, *Mcam*, and had weak but detectable signal for *Cd34* (***Figure 6A***). Myh11CreER^{T2} sorted cells showed expression of smooth muscle actin (*Acta2*), the alpha subunit of the L-type voltage-gated $Ca^{2+}$ channel *Cacna1c* (Cav1.2), *Desmin* (*Des*), *Mcam*, and *Vimentin* (*Vim*, ***Figure 6B***). In addition to the genes expressed in Myh11CreER^{T2} recombined cells, *Cdh5*, *Cd34*, and *Cspg4* (also known as *Ng2*) were also detected in cells sorted from PdgfrβCreER^{T2} IALVs (***Figure 6C***). As expected, the GFP⁺ cells sorted from PdgfrαCreER™ IALVs expressed mRNA for *Cd34*, weak signal for *Cspg4*, and *Vimentin*, but not *Desmin*, *Acta2*, nor the pericyte marker *Mcam* (***Figure 6D***). *Cacna1c* was expressed in cells FACS purified from both PdgfrβCreER^{T2} and Myh11CreER^{T2} IALVs and sorted cells from PdgfrαCreER™ IALVs without any evidence that *Myh11*-expressing muscle cells contaminated the latter. These findings confirmed the separate cell populations achieved with PdgfrαCreER- and Myh11CreER^{T2}-mediated recombination, at least as it pertains to the *Rosa26^{mTmG}* reporter. These findings were largely validated by our scRNASeq dataset. *Cdh5* (***Figure 6E***) and *Nos3* (***Figure 6F***) were almost exclusively expressed in the LEC clusters, while *Acta2* (***Figure 6G***) was highly expressed in the LMC cluster. We also observed that *Cacna1c* was highly expressed in the LMCs (***Figure 6H***) and some AdvCs. *Cd34* was widely expressed in AdvCs matching our immunofluorescence data. *Cd34* expression was also seen in LECs (***Figure 6I***) although we did not observe a signal in LECs in our earlier immunofluorescence staining (***Figure 3***). *Cspg4* was observed in a minor population of AdvCs (***Figure 6J***). The intermediate filament *Vim* (***Figure 6K***) was ubiquitously expressed across all clusters expressed, but *Des* was primarily expressed in LMCs and some subsets of AdvCs (***Figure 6K, L***). The endothelial and pericyte marker *Mcam* (also referred to as *Cd146*) was expressed in LECs and LMCs but was largely absent in AdvCs (***Figure 6M***). We followed up the identification of *Cacna1c* expression in the PdgfrαCreER™ sorted cell population by assessing the expression of other genes involved in either electrical conduction (*Gjc1*) (***Figure 6N***) or pacemaking (*Ano1*) (***Figure 6O***) of IALVs. Expression of *Ano1* and *Gjc1* was observed in PdgfrαCreER™ *Rosa26^{mTmG}* FACS-purified cells (***Figure 6P***).

## Inducible deletion of either *Cacna1c*, *Ano1*, or *Gjc1* with PdgfrαCreER™ did not affect cLV pacemaking

The expression of the genes critically involved in cLV function—*Cacna1c*, *Ano1*, and *Gjc1*—in the PdgfrαCreER™-*Rosa26^{mTmG}* purified cells and scRNAseq data prompted us to generate PdgfrαCreER™-*Ano1^{fl/fl}*, PdgfrαCreER™-*Gjc1^{fl/fl}*, and PdgfrαCreER™-*Cacna1c^{fl/fl}* mice for contractile tests. We isolated popliteal cLVs and tested their pacemaker and contractile function in response to a physiological pressure range of 0.5–10 cmH₂O, under normal conditions. However, we did not detect any significant differences in pacemaking or contractile function as assessed by contraction frequency, ejection fraction, and vessel tone in popliteal cLVs studied from PdgfrαCreER™-*Ano1^{fl/fl}* mice (***Figure 7A–C***) or PdgfrαCreER™-*Gjc1^{fl/fl}* mice (***Figure 7D–F***). There was no difference in contraction frequency of cLVs from PdgfrαCreER™-*Cacna1c^{fl/fl}* mice compared to floxed control mice; however, we noted a mild but statistically significant increase in ejection fraction at the lowest pressure, 0.5 cmH₂O (***Figure 7H***). Additionally, vessels isolated from PdgfrαCreER™-*Cacna1c^{fl/fl}* mice also had a statistically significant

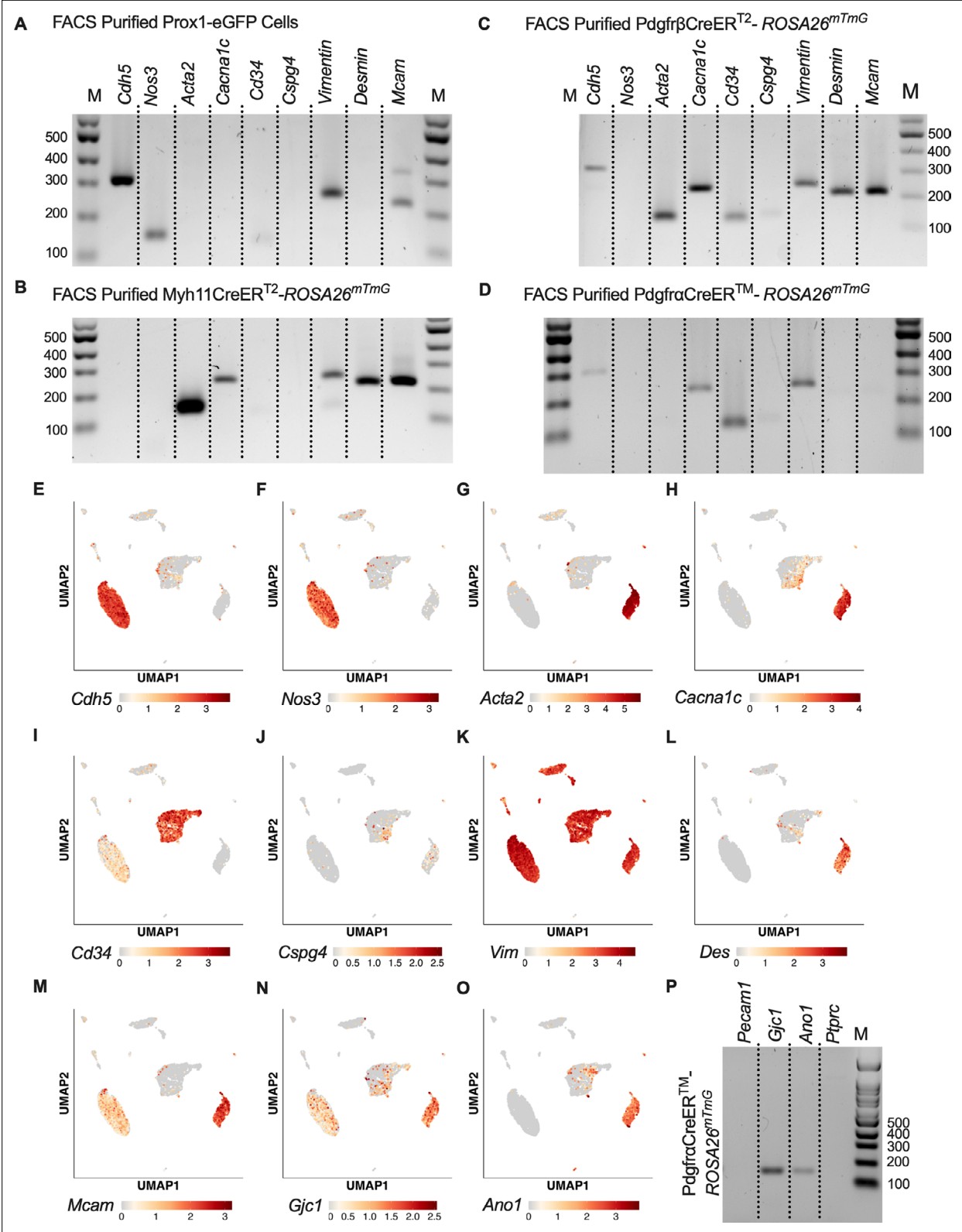

**Figure 6.** RT-PCR Profiling of FACS-purified cells from inducible Cre-*Rosa26^{mTmG}*. Expanded RT-PCR profiling of genes to discriminate lymphatic endothelial cells (LECs), lymphatic muscle cells (LMCs), and other cell types in our GFP⁺ sorted cells from Prox1-eGFP (**A**), Myh11CreER^T2^-*Rosa26^{mTmG}* (**B**), PdgfrβCreER^T2^-*Rosa26^{mTmG}* (**C**), and PdgfrαCreER^TM^-*Rosa26^{mTmG}* (**D**). Feature plots for the genes assessed in A-D in our IALV scRNAseq analysis confirmed those results (**E-M**). In addition to a population of AdvCs expressing *Cacna1c*, we also noted expression of *Gjc1* (**N**), which was also observed

*Figure 6 continued on next page*

*Figure 6 continued*

in LECs, and *Ano1* (**O**) in the AdvC clusters. We confirmed this expression using GFP⁺ cells sorted from PdgfrαCreER™-*Rosa26ᵐᵀᵐᴳ* IALVs for RT-PCR (**P**) and ruled out hematopoietic or LEC contamination. All RT-PCRs were performed two to four times for each gene over each sorted cell population collected from different mice.

The online version of this article includes the following video, source data, and figure supplement(s) for figure 6:

**Source data 1.** This file contains the RT-PCR gel electrophoresis data for *Figure 6* without markup.

**Source data 2.** This file contains the RT-PCR gel electrophoresis data for *Figure 6* without markup.

**Figure supplement 1.** PDGFRα AdvCs include multipotent cell.

**Figure supplement 1—source data 1.** This file contains the RT-PCR gel electrophoresis data for *Figure 6—figure supplement 1*.

**Figure supplement 1—source data 2.** This file contains the RT-PCR gel electrophoresis data for *Figure 6—figure supplement 1* without markup.

**Figure 6—video 1.** Lymphatic PDGFRα⁺ AdvCs co-stain with the stem cell marker LY6A.

https://elifesciences.org/articles/90679/figures#fig6video1

increase in vessel tone (*Figure 7I*) noted at the two-way level although we did not resolve significance at any specific pressure with this sample. No differences in normalized contraction amplitude, fractional pump flow, or diastolic diameter were observed (*Figure 7—figure supplement 1*). In total, despite the presence of transcript for these critical genes in *Pdgfra* expresing cells, PdgfrαCreER™-mediated deletion of *Gjc1*, *Cacna1c*, or *Ano1* failed to recapitulate previous reports of the significant contractile defects using the Myh11CreER™² line to delete the same genes (*Castorena-Gonzalez et al., 2018b*; *Zawieja et al., 2019*; *To et al., 2020*; *Davis et al., 2022*).

## PDGFRα⁺ adventitial fibroblasts express markers associated with multipotency

Despite the lack of cLV pacemaking deficits in the PdgfrαCreER™ genetic knockout lines, we were curious to discern further insight into the role or function of the PDGFRα⁺ CD34⁺ cells since they comprise a significant portion of the lymphatic cLV wall. We performed RT-PCR on FACS-purified cells from Prox1-eGFP, Myh11CreER™²-*Rosa26ᵐᵀᵐᴳ*, and PdgfrαCreER™-*Rosa26ᵐᵀᵐᴳ* IALVs for multipotency markers including Krüppel-like factor 4 (*Klf4*), lymphocyte antigen 6 family member A (*Ly6a*) (also referred to as stem cell antigen 1, *Sca1*), and *Gli1*, with *Cd34* and *Pdgfra* used to assess purity. Recombined (GFP⁺) cells from Myh11CreER™²-*Rosa26ᵐᵀᵐᴳ* had weak expression of *Klf4* and *Gli1* but were negative for *Ly6a* (*Figure 6—figure supplement 1A*). PdgfrαCreER™ recombined cells strongly expressed *Klf4*, *Ly6a*, and *Gli1* (*Figure 6—figure supplement 1A*). LECs sorted from Prox1-eGFP IALVs were positive for *Klf4*, weak for *Ly6a*, and positive for *Cd34* but negative for *Gli1* and *Pdgfra* (*Figure 6—figure supplement 1B*). The unrecombined population (tdTomato⁺) cells in the Myh11CreER™²- *Rosa-26ᵐᵀᵐᴳ* IALVs (*Figure 6—figure supplement 1B*) showed expression for all the markers as expected. PdgfrαCreER™ *recombined cells* also expressed the mesenchymal stromal cell markers *Itgb1*, *Eng*, and *Cd44* (*Figure 6—figure supplement 1C*, positive control in 9D). However, expression of these genes was not homogenous across all the AdvCs population based on our scRNAseq analysis (*Figure 6—figure supplement 1E–J*). We performed immunofluorescence staining for one of these multipotent markers, LY6A (*Figure 6—figure supplement 1K*) in the adventitial cells with PDGFRα (*Figure 6—figure supplement 1L*) and counter staining for LMCs with MYH11 (*Figure 6—figure supplement 1M*). The morphology and staining pattern of LY6A overlapped significantly with PDGFRα staining and not MYH11 staining (*Figure 6—figure supplement 1N*, *Figure 6—video 1*).

## Optogenetic stimulation of inducible Cre-driven channel rhodopsin 2

We next used optogenetic methods to test whether the cell populations recombined by either Ckit-CreER™², PdgfrαCreER™, or Myh11CreER™² could elicit a coordinated contraction. The ChR2-tdTomato construct appeared more sensitive to recombination than *Rosa26ᵐᵀᵐᴳ*, in some cases resulting in LMC expression of ChR2-tdTomato in PdgfrαCreER™ and CkitCreER™² popliteal cLVs based on cell morphology. Care was taken to image each vessel for tdTomato (*Figure 8A, C, E*) prior to stimulation at its respective sites under brightfield conditions for diameter tracking (*Figure 8B, D, F*) to ensure fidelity of the cell types and morphologies observed in *Figures 3 and 4*. As with *Rosa26ᵐᵀᵐᴳ*, Ckit-CreER™² drove the ChR2-tdTomato expression primarily in large ovoid cells found on the adventitia

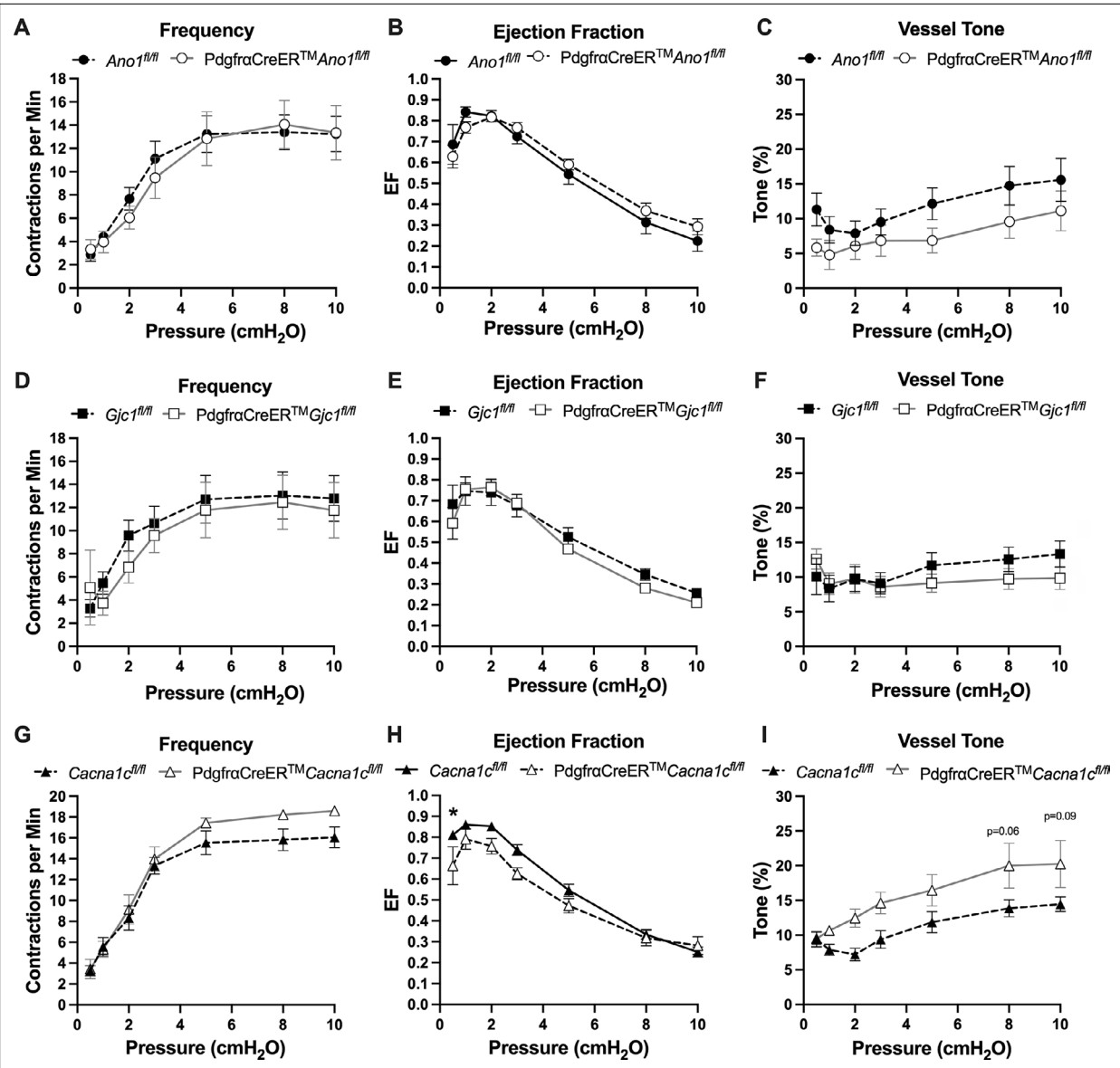

**Figure 7.** Isobaric contractile assessment of popliteal collecting lymphatic vessel (cLV) from PdgfrαCreER™-driven deletion of *Ano1*, *GJC1*, and *Cacna1c*. Summary of the contractile parameters recorded from popliteal cLVs in PdgfrαCreER™-*Ano1*^fl/fl, PdgfrαCreER™-*Gjc1*^fl/fl mice, PdgfrαCreER™-*Cacna1c*^fl/fl mice. Contraction frequency (**A**, **D**, **G**), ejection fraction (**B**, **E**, **H**), and vessel tone (**C**, **F**, **I**) were assessed. No statistically significant differences were observed in cLVs isolated from PdgfrαCreER™-*Ano1*^fl/fl and PdgfrαCreER™-*Gjc1*^fl/fl mice across these three parameters. Mean and SEM shown, *n* = 6 popliteal vessels from three mice PdgfrαCreER™-*Ano1*^fl/fl mice and *n* = 9 popliteal vessels from six mice *Ano1*^fl/fl mice. Mean and SEM shown, *n* = 5 popliteal vessels from three mice PdgfrαCreER™-*GJC1*^fl/fl mice and *n* = 11 popliteal vessels from eight mice *GJC1*^fl/fl mice. Mean and SEM shown, *n* = 6 popliteal vessels from three mice PdgfrαCreER™-*Cacna1c*^fl/fl mice and *n* = 8 popliteal vessels from eight mice *Cacna1c*^fl/fl mice. The contractile data from control *Cacna1c*^fl/fl vessels is a subset of previously published data that was separated by sex (*Davis et al., 2022*) while they are combined here. * denotes significance at p < 0.05 which 0.10 > p > 0.05 are reported as text. Normalized contraction amplitude, fractional pump flow, end diastolic diameter can be found in *Figure 7—figure supplement 1*.

The online version of this article includes the following figure supplement(s) for figure 7:

**Figure supplement 1.** Contractile indices from isobaric myography on collecting lymphatic vessels (cLVs) from PdgfrαCreER™-driven deletion of *Ano1*, *GJC1*, and *Cacna1c*.

of the vessel. Cells were stimulated by positioning an optical laser fiber (tip diameter 2–3 μm) near a ChR2+ cell, with an illumination field of 10–50 μm. Localized photo-stimulation of these cells did not initiate coordinated contractions (*Figure 8G–J, S*). Similarly, photo-stimulation of ChR2-tdTomato expressing cells driven by PdgfrαCreER™ failed to initiate a coordinated contraction (*Figure 8K–N,*

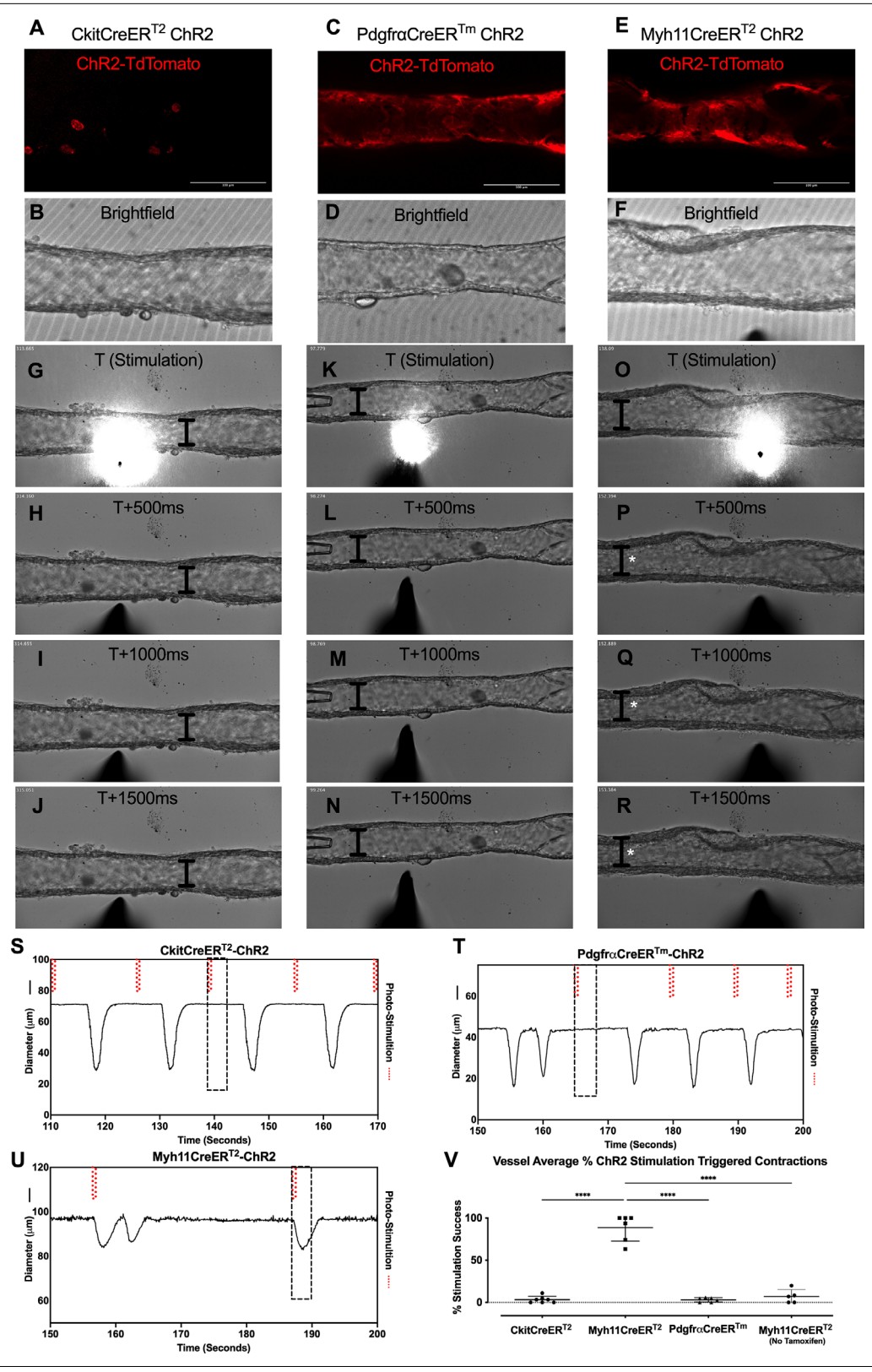

**Figure 8.** ChR2-mediated depolarization only in lymphatic muscle cells (LMCs) triggers contraction. Representative max projections of tdTomato-ChR2 signal in popliteal collecting lymphatic vessels (cLVs) isolated from CkitCreER$^{T2}$-ChR2-tdTomato (**A**), PdgfrαCreER$^{TM}$-ChR2-tdTomato (**C**), and Myh11CreER$^{T2}$-ChR2-tdTomato (**E**) with their corresponding brightfield image (**B**, **D**, **F**), respectively. Time-lapse brightfield images every 0.5 s starting at

*Figure 8 continued on next page*

*Figure 8 continued*

stimulation $t = 0$ for CkitCreER$^{T2}$-ChR2-tdTomato (**G–J**), PdgfrαCreER$^{TM}$-ChR2-tdTomato (**K–N**), and Myh11CreER$^{T2}$-ChR2-tdTomato (**O–R**). The I bar denotes the inner diameter at $t = 0$ over time, and white asterisks denote the contraction. Representative diameter trace for the popliteal cLV demonstrates spontaneous contractions with the dotted boxes indicating the optical stimulation event in the respective brightfield images of the time lapse images. Isolated cLVs from CkitCreER$^{T2}$-ChR2-tdTomato (**S**), PdgfrαCreER$^{TM}$-ChR2-tdTomato (**T**), and Myh11CreER$^{T2}$-ChR2-tdTomato (**U**) were stimulated with light pulses (red dashed lines) and the summation of contraction triggering for each genotype (**V**). Mean and SEM are shown, **** denotes $p < 0.0001$. Contraction recorded from at least six popliteal cLVs from 3 mice per genotype.

The online version of this article includes the following video(s) for figure 8:

**Figure 8—video 1.** Brightfield video of optogenetic stimulation of IALVs from CkitCreER$^{T2}$-ChR2 mice.
https://elifesciences.org/articles/90679/figures#fig8video1

**Figure 8—video 2.** Brightfield video of optogenetic stimulation of IALVs from PdgfrαCreER-ChR2 mice.
https://elifesciences.org/articles/90679/figures#fig8video2

**Figure 8—video 3.** Brightfield video of optogenetic stimulation of IALVs from Myh11CreER$^{T2}$-ChR2 mice.
https://elifesciences.org/articles/90679/figures#fig8video3

---

**T**). In contrast, localized photo-stimulation of LMCs, using Myh11CreER$^{T2}$ to express Chr2-tdTomato, resulted in a propagated contraction in the popliteal vessel (*Figure 8O–R, U*). In total, only 3.25% of photo-stimulation events for CkitCreER$^{T2}$-ChR2-TdTomato and 3.03% of photo-stimulation events for PdgfrαCreER$^{TM}$-ChR2-tdTomato were associated with a contraction, while 88.5% of photo-stimulation events for Myh11CreER$^{T2}$-ChR2-tdTomato induced contractions (*Figure 8V*). The optogenetic triggering of contractions observed in PdgfrαCreER$^{TM}$-ChR2-tdTomato and CkitCreER$^{T2}$-ChR2-TdTomato vessels is likely due to the happenstance of spontaneous contractions occurring during the time and proximity of optogenetic stimulation (see Methods). As a control, we also used non-induced (no tamoxifen) Myh11CreER$^{T2}$-ChR2-tdTomato cLVs and contractions were associated with only 7% of photo-stimulation events, in line with the PdgfrαCreER$^{TM}$ and CkitCreER$^{T2}$ results (*Figure 8V*). As mast cells are not ascribed to any tissue-specific pacemaking behavior, these similar low percentages observed between these three groups are suggestive of random coincidence. Brightfield videos of the photo-stimulation and representative traces for CkitCreER$^{T2}$-ChR2-TdTomato, PdgfrαCreER$^{TM}$-ChR2-tdTomato, and Myh11CreER$^{T2}$-ChR2-tdTomato are provided in *Figure 8—video 1*; *Figure 8—video 2*; *Figure 8—video 3*.

## Confocal Ca$^{2+}$ imaging of GCaMP6f expression driven by CkitCreER$^{T2}$, PdgfrαCreER$^{TM}$, and Myh11CreER$^{T2}$ over the lymphatic contraction cycle

Subcellular calcium transients are observed in many pacemaker cells. We imaged IALVs from CkitCreER$^{T2}$-GCaMP6f mice, which primarily resulted in expression of GCaMP6f in the large ovoid cells in the adventitia (*Figure 9A*), although we occasionally observed GCaMP6f expression in both LEC and LMCs (*Figure 9A*) as depicted in the maximum projection of the acquisition period (*Figure 9—video 1*) and the spatio-temporal maps (STMs). The aberrant expressions of GCaMP6f in cells that demonstrated the typical cobblestone morphology of LECs or the circumferential LMCs that exhibited Ca$^{2+}$ flashes and diastolic Ca$^{2+}$ transients (*Figure 9D, E*, green arrows) prior to contraction were not included in the CkitCreER$^{T2}$-GCaMP6f analysis. Of 39 CkitCreER$^{T2}$-GCaMP6f cells analyzed, only 1 CkitCreER$^{T2}$-GCaMP6f cell exhibited a spontaneous Ca$^{2+}$ transient during the recording period (*Figure 9B, C*, Cell 7). However, the Ca$^{2+}$ transient in that cell did not align temporally with the 'Ca$^{2+}$ flash' of the LMC with incidental GCaMP6f expression (*Figure 9C, D*). Despite the lack of Ca$^{2+}$ transients under the baseline conditions throughout the IALV contraction cycle, many CkitCreER$^{T2}$-GCaMP6f cells exhibited a robust and prolonged Ca$^{2+}$ event in response to stimulation with the mast cell activator compound 48–80 (*Figure 9F–H*). Notably, the Ca$^{2+}$ events in the ovoid cells elicited by administration of compound 48–80 did not acutely alter the LMC Ca$^{2+}$ activity (*Figure 9I, J*). Similarly, the majority of PdgfrαCreER$^{TM}$-GCaMP6f expressing cells also largely lacked Ca$^{2+}$ transients and resulted in incidental LMC GCaMP6f expression (*Figure 10B*, *Figure 10—video 1*). Some cells exhibited high basal Ca$^{2+}$ levels (*Figure 10A, D*) sustained throughout the recording, but without oscillations (*Figure 10B, C*).

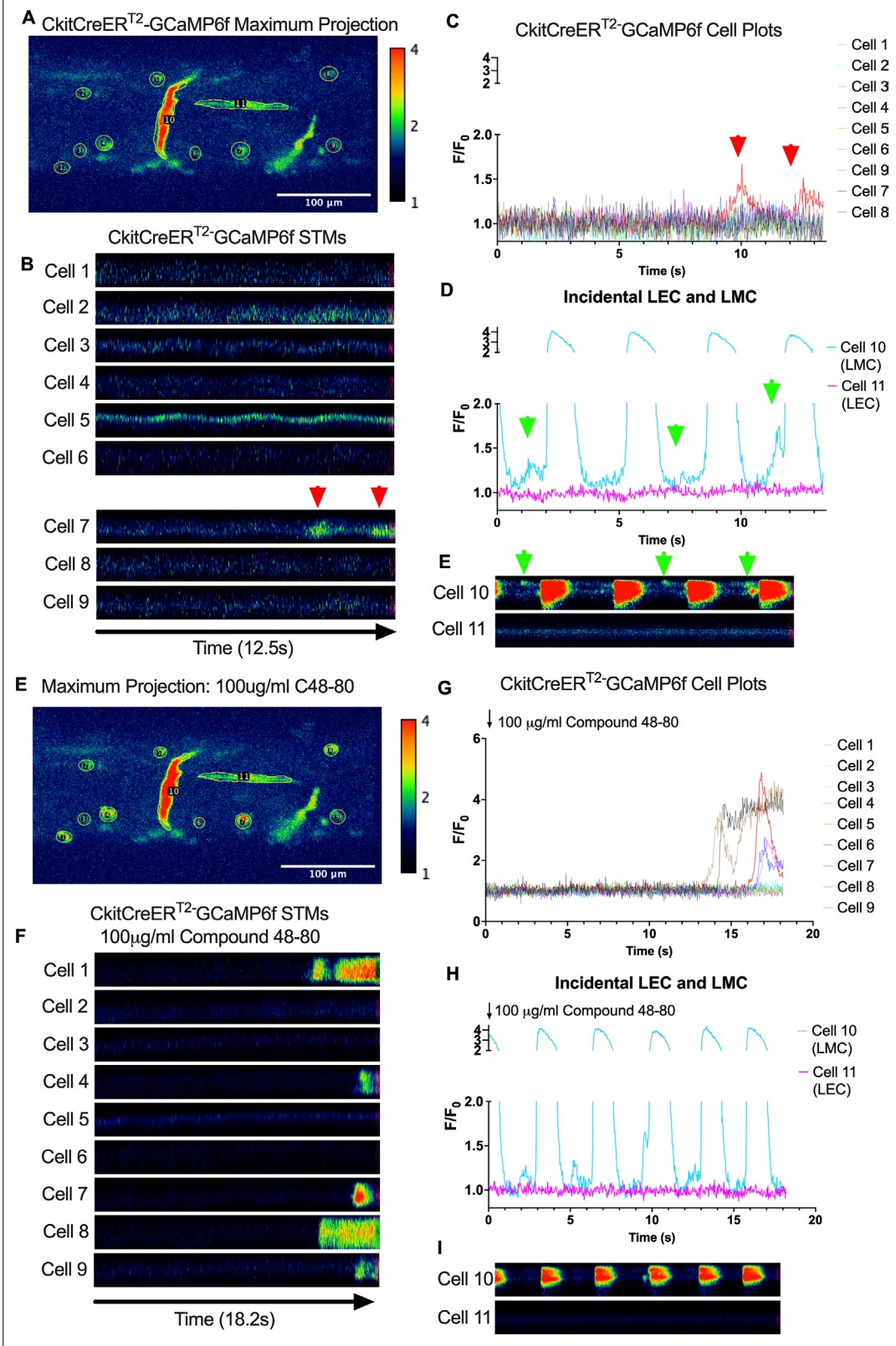

**Figure 9.** CkitCreER[T2] drives GCaMP6f expression primarily in mast cells in mouse IALVs. Representative max projection (**A**) of GCaMP6f signal over time in an IALV isolated from a CkitCreER[T2]-GCaMP6f mouse with ROI indicated around individual cells, primarily large ovoid cells, but also including a circumferential lymphatic muscle cell (LMC) (cell 10) and a horizontal lymphatic endothelial cell (LEC) (cell 11). Of cells 1–9, only cell 7 had any Ca[2+]

*Figure 9 continued on next page*

*Figure 9 continued*

activity (red arrows) during the recording time as indicated by the spatio-temporal maps (STMs) from each ROI (**B**) and their normalized $F/F_0$ plots in (**C**). In contrast, the LMC in ROI 10 had both rhythmic global $Ca^{2+}$ events (**D**) that spanned the cell axis (vertical axis) in the STM (**E**) in addition to localized $Ca^{2+}$ events intervening the time between global events (green arrows). Representative max projection of GCaMP6f signal over time after stimulation with C48–80 (**F**) with many large ovoid cells displaying long-lasting global $Ca^{2+}$ events (**G**, **H**) while not immediately affecting the LMC $Ca^{2+}$ dynamics (**I**). Calcium recordings were made in $n$ = 6 IALVs from four mice.

The online version of this article includes the following video for figure 9:

**Figure 9—video 1.** Calcium imaging in an IALV isolated from a CkitCreER^T2^-GCaMP6f mouse.
https://elifesciences.org/articles/90679/figures#fig9video1

In contrast, spurious GCaMP6f expression in a circumferentially oriented LMC displayed $Ca^{2+}$ flashes associated with contraction (*Figure 10B, C*). Of the 21 PdgfrαCreER^TM^ -GCaMP6f cells assessed, only 3 exhibited $Ca^{2+}$ transients, and those were singular events contained within a single cell within the 20-s imaging period (*Figure 10E, F*). The lack of either global or consistent $Ca^{2+}$ transients within either CkitCreER^T2^-GCaMP6f or PdgfrαCreER^TM^-GCaMP6f IALVs was in stark contrast to $Ca^{2+}$ imaging of Myh11CreER^T2^-GCaMP6f IALVs. Myh11CreER^T2^ drove GCaMP6f expression in nearly all circumferential LMCs (*Figure 11A*), which exhibited global and nearly synchronous $Ca^{2+}$ flashes in 100% of the analyzed cells (*Figure 11B, C*). Additionally, non-synchronous stochastic and localized $Ca^{2+}$ transients were commonly observed in the LMCs during diastole (*Figure 11D, E*, *Figure 11—video 1*). Many LMCs exhibited $Ca^{2+}$ transients during each diastolic period while other LMCs displayed few $Ca^{2+}$ transients or lacked diastolic $Ca^{2+}$ transients during the recording period (*Figure 11B*). In aggregate, only 1 of 39 CkitCreER^T2^-GCaMP6f cells and 3 of 21 PdgfrαCreER^TM^-GCaMP6f cells displayed a $Ca^{2+}$ transient during recording, while 20 of 43 LMCs displayed at least one diastolic transient apart from 43 of 43 LMCs with global $Ca^{2+}$ flashes.

## Pressure dependency of subcellular $Ca^{2+}$ transients in LMCs

We next sought to test whether diastolic $Ca^{2+}$ transients were pressure-dependent, given that cLVs exhibit pressure-dependent chronotropy (*Zawieja et al., 2019*). GCaMP6f-expressing LMCs were studied at intraluminal pressures of 0.5–5 cmH$_2$O in the presence of nifedipine, which blocked the $Ca^{2+}$ flashes but not local $Ca^{2+}$ transients (*Figure 12A*). As intra-luminal pressure was increased, there was a marked increase in the occurrence of $Ca^{2+}$ transients (*Figure 12B*, *Figure 12—videos 1–3*). These calcium transients were converted into particles (PTCLs) for further analysis as previously described (*Drumm et al., 2019b*). Activity maps of $Ca^{2+}$ PTCL activity were generated (*Figure 12C*) and PTCL area (*Figure 12D*) and frequency were determined at each pressure (*Figure 12E*). The maps show that as pressure increased, the area of the LMC layer displaying a $Ca^{2+}$ transient increased (as evident by the increase in PTCL area) as did the distribution of $Ca^{2+}$ PTCLs across the LMC layer (*Figure 12C*). Across 11 experiments, the area of the field of view activated by PTCLs/frame increased from 73.2 ± 17.7 μm$^2$/frame at 0.5 cmH$_2$O to 108.6 ± 20.5 μm$^2$/frame at 2 cmH$_2$O and was further enhanced to 139.2 ± 26.9 μm$^2$/frame at 5 cmH$_2$O (*Figure 12F*). The number of PTCLs per frame also increased with pressure, from 2.9 ± 0.4 at 0.5 cmH$_2$O to 4.1 ± 0.5 and 5.2 ± 0.6 PTCL/frame at 2 and 5 cmH$_2$O, respectively (*Figure 12G*).

## Contraction frequency is set by the diastolic depolarization rate

To assess how pressure regulates LMC membrane potential, we first recorded membrane potential in cells exhibiting action potentials (APs) using a microelectrode filled with biocytin-AF488 to label the impaled cell. In each case ($n$ = 3 IALVs), the labeled cell was an LMC wrapping circumferentially around the vessel (*Figure 13A, B*), and as these recordings were made over the course of many minutes, the direct neighboring circumferential LMCs also exhibited fluorescence, albeit weaker in intensity, as expected for cells coupled by gap junctions (*Figure 13A*). In all the recorded cells exhibiting APs, we noted a diastolic depolarization preceding the sharp upstroke achieved once threshold was met at each pressure (*Figure 13C*). The AP frequency and rate of the diastolic depolarization increased with pressure (*Figure 13D, E*). Linear regression of a plot of each AP frequency and diastolic depolarization rate at each pressure demonstrated a tight association between the two parameters. However, we did not observe a significant effect of pressure on minimum membrane potential (*Figure 13G*), threshold

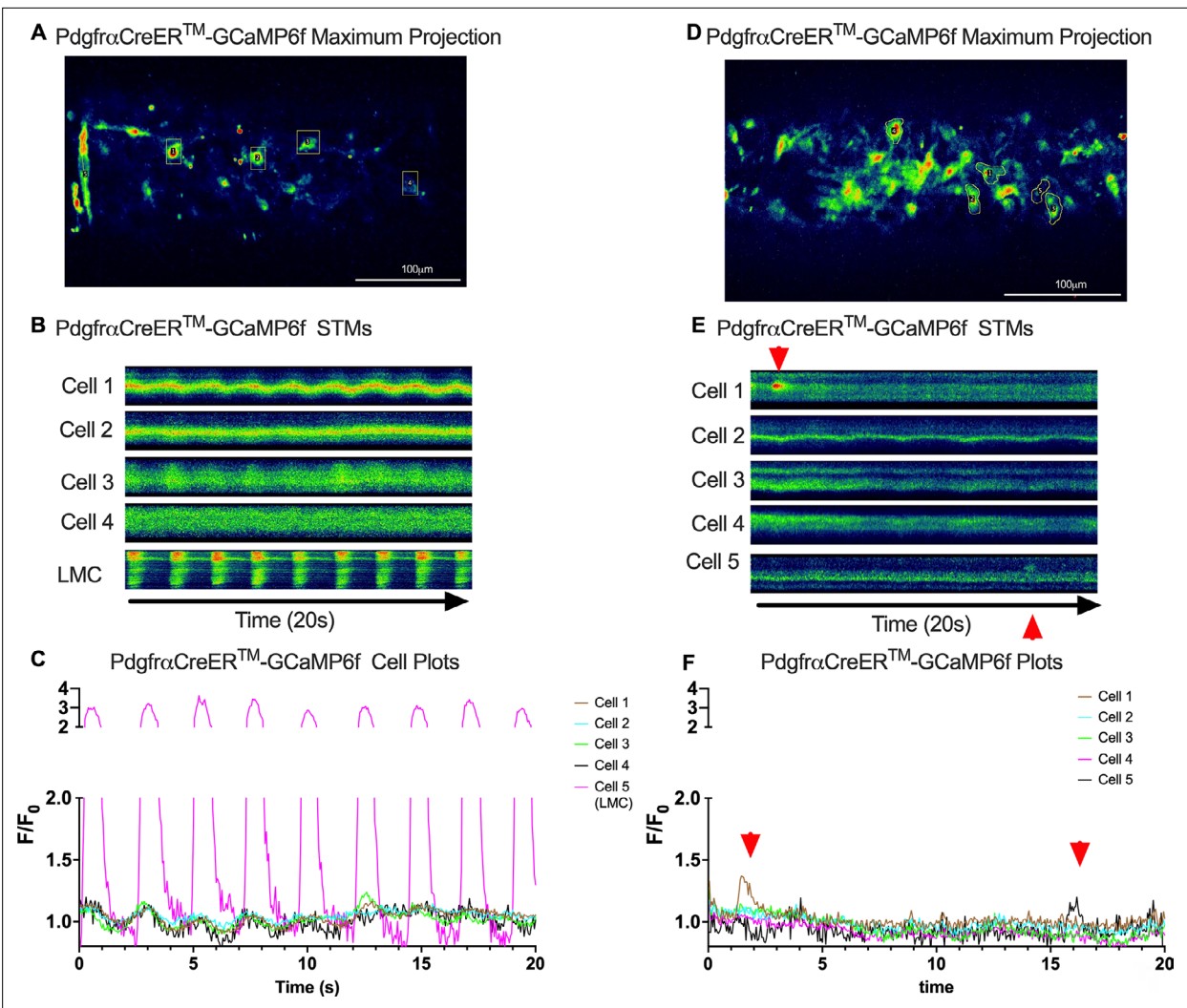

**Figure 10.** Lack of coordinated Ca²⁺ activity across contraction cycle in PDGFRα cells. Representative max projections of GCaMP6f signal over time in IALVs isolated from PdgfrαCreERᵀᴹ-GCaMP6f mice (**A**, **D**). ROIs were made around cells and GCaMP6f recorded over time to generate the corresponding spatio-temporal maps (STMs) (**B**, **E**) for each cell and plots (**C**, **F**), respectively. Once again, incidental recombination occurred in a lymphatic muscle cell (LMC) which displayed rhythmic Ca²⁺ flashes (**C**) while the slight undulation in the other cells is due to movement artifact (**B**). Red arrows indicate the limited local Ca²⁺ activity observed in two cells from a PdgfrαCreERᵀᴹ-GCaMP6f IALV. Calcium recordings were made in $n$ = 6 IALVs from four mice.

The online version of this article includes the following video for figure 10:

**Figure 10—video 1.** Calcium imaging in an IALV isolated from a PdgfrαCreER-GCaMP6f mouse.

https://elifesciences.org/articles/90679/figures#fig10video1

potential (*Figure 13H*), AP upstroke (*Figure 13I*), AP peak potential (*Figure 13J*), plateau potential (*Figure 13K*), or the time spent over threshold (*Figure 13L*).

## Discussion

The identification of the cellular origin and signaling mechanisms underlying cLV pacemaking will reveal novel targets for pharmacological intervention in treating lymphedema and the associated lymphatic contractile dysfunction. In this study, we tested proposed pacemaker cell types based on three parameters: (1) that the pacemaker cells are located along the entire length of the cLV, to accommodate spontaneous contractions and coordinated electrical conduction; (2) that depolarization of the pacemaker cell can drive a coordinated and propagated contraction of the vessel; and (3) that the presence of Ca²⁺ transients precedes or coincides with contraction, as commonly observed

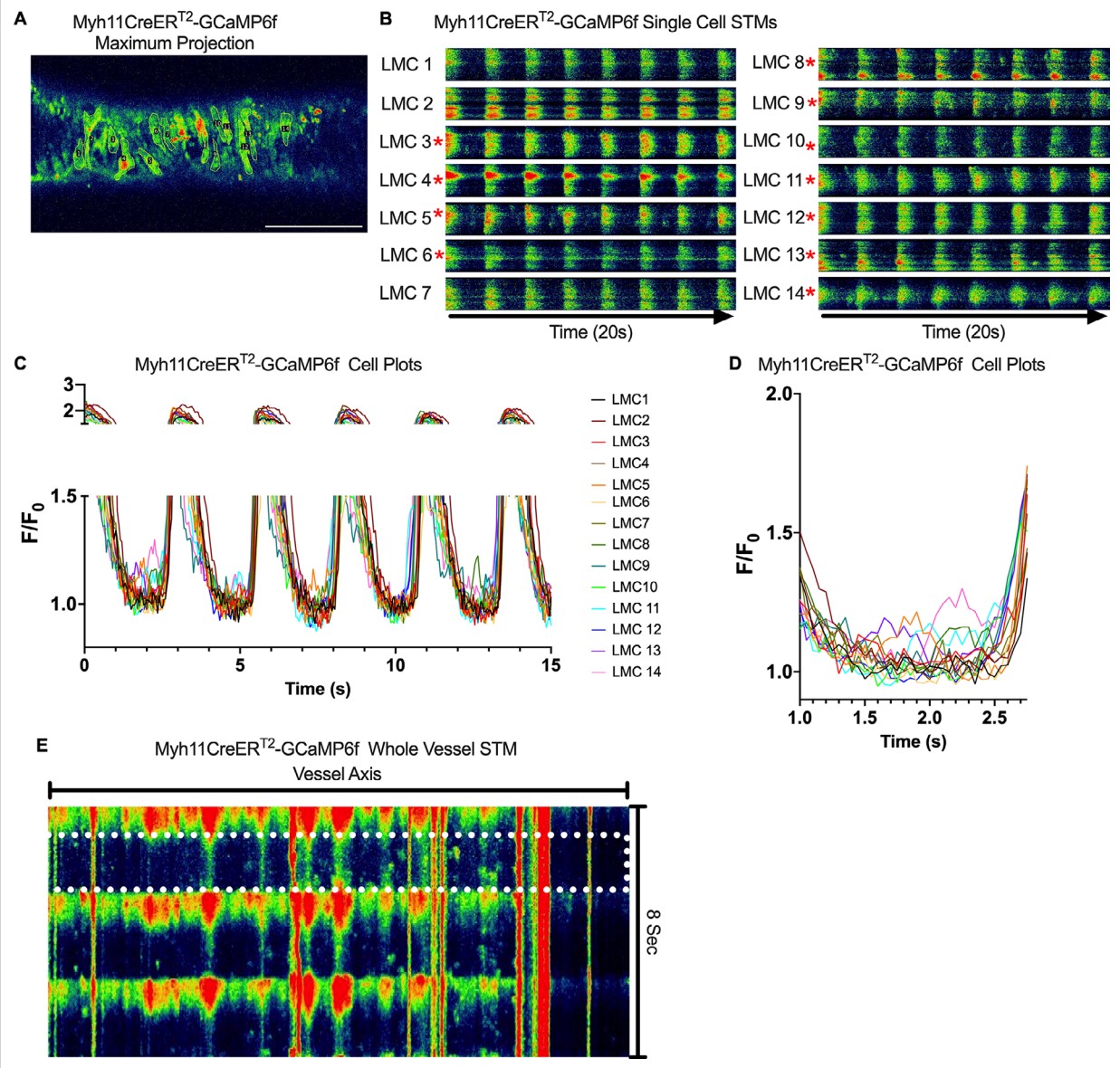

**Figure 11.** Heterogeneous diastolic $Ca^{2+}$ transient activity in lymphatic muscle cells (LMCs). Representative max projections of GCaMP6f signal over time in an IALVs isolated from Myh11CreER$^{T2}$-GCaMP6f mice (**A**). LMCs were outlined with ROIs to assess GCaMp6F signal over time. Rhythmic global flashes (**B**) were entrained across all the LMCs in the FOV (**C**) with many cells exhibiting diastolic $Ca^{2+}$ release events. Cells exhibiting at least one diastolic $Ca^{2+}$ event, within the context of our focal plane constraints, over the recorded time were denoted by the red asterisks. The plot in (**D**) magnifies the first diastolic period, seconds 1–3 of C, to assist in visualizing the lack of coordination of the diastolic events. (**E**) Max projection of the pseudo-linescan analysis across the axis of the vessel to highlight diastolic $Ca^{2+}$ transients in all cells in the field of view and their lack of coordination across the cells (x-axis). The white dotted box shows the first diastolic period plotted in (**D**). Representative images from calcium recordings from n = 4 IALVs from four mice.

The online version of this article includes the following video for figure 11:

**Figure 11—video 1.** Calcium imaging in an IALV isolated from a Myh11CreER$^{T2}$-GCaMP6f mouse.
https://elifesciences.org/articles/90679/figures#fig11video1

in other pacemaker cells. We used confocal microscopy and a combination of immunofluorescence and fluorescent reporters under the control of various inducible Cres to identify and target both muscle and non-muscle cells, previously labeled as ICLCs, which co-stain for the markers CD34 and PDGFRα. Our cell characterizations were supplemented by scRNAseq analysis of isolated and cleaned mouse IALVs which supported our finding of three major cell types including LECs, LMCs, and AdvCs,

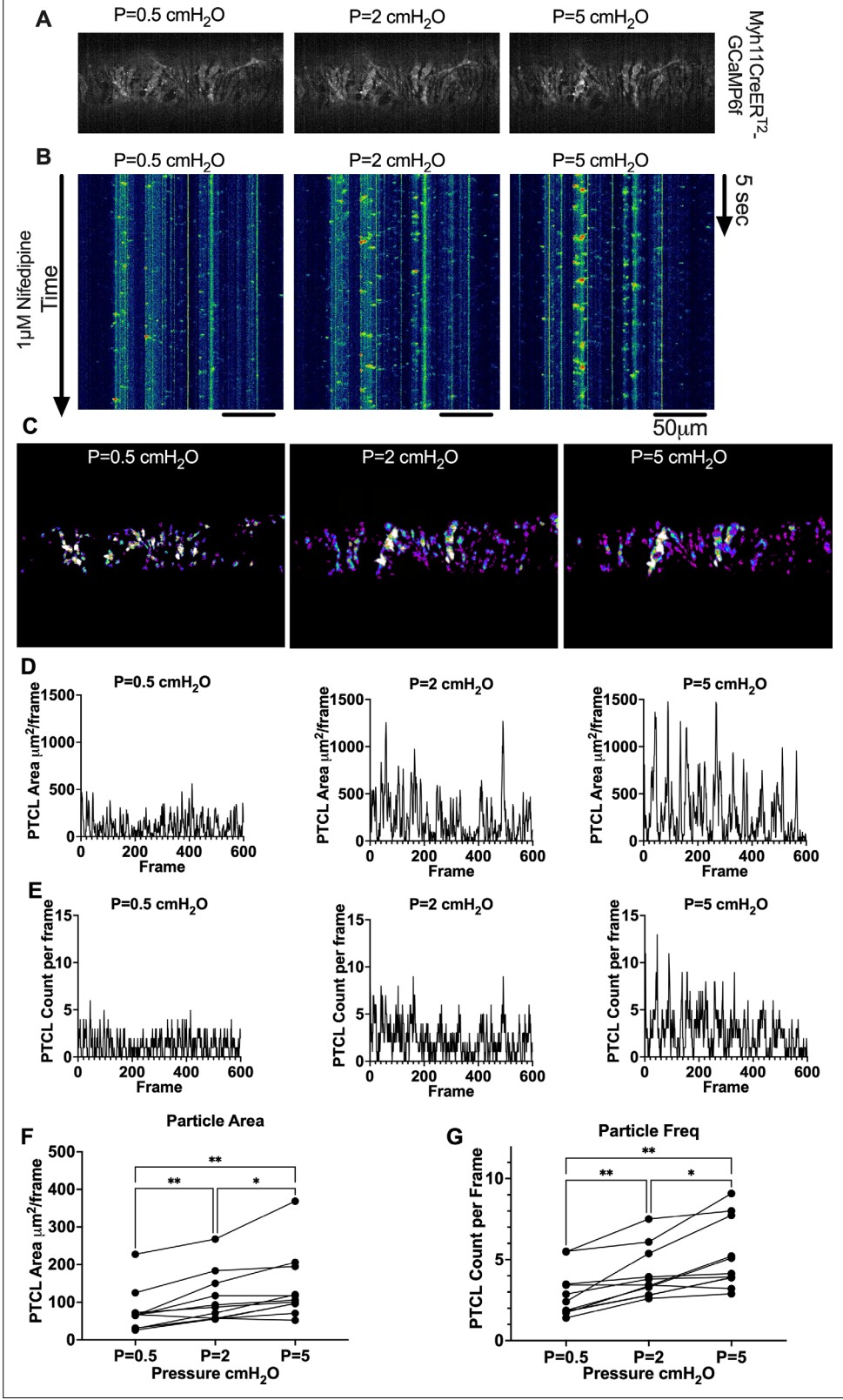

**Figure 12.** Pressure dependency of mouse lymphatic muscle cell (LMC) diastolic Ca²⁺ transients. Representative max projection of GCaMP6f signal over 20 s in an IALVs isolated from Myh11CreER^T2^-GCaMP6f mice in the presence of the L-type blocker nifedipine (1 μM) (**A**) and pressurized to 0.5, 2, and 5 cmH₂O. The local diastolic Ca²⁺ transients persist in the presence of nifedipine and increase with increasing pressure as demonstrated in the

*Figure 12 continued on next page*

*Figure 12 continued*

whole vessel spatio-temporal maps (STMs) (**B**). Particle occurrence maps highlight the $Ca^{2+}$ activity in each LMC as pressure is raised (**C**). Representative particle analysis plots for particle area (**D**) and particle counts/frame at each pressure (**E**). Summary files for particle area (**F**) and count/frame (**G**). * denotes $p < 0.05$, Mean and SEM shown with $n = 12$ separate IALVs from 8 Myh11CreER$^{T2}$-GCaMP6f.

The online version of this article includes the following video(s) for figure 12:

**Figure 12—video 1.** Imaging subcellular calcium oscillations, at a pressure of 0.5 cmH$_2$O, in an IALV isolated from a Myh11CreER$^{T2}$-GCaMP6f mouse.

https://elifesciences.org/articles/90679/figures#fig12video1

**Figure 12—video 2.** Imaging subcellular calcium oscillations, at a pressure of 2.0 cmH$_2$O, in an IALV isolated from a Myh11CreER$^{T2}$-GCaMP6f mouse.

https://elifesciences.org/articles/90679/figures#fig12video2

**Figure 12—video 3.** Imaging subcellular calcium oscillations, at a pressure of 5.0 cmH$_2$O, in an IALV isolated from a Myh11CreER$^{T2}$-GCaMP6f mouse.

https://elifesciences.org/articles/90679/figures#fig12video3

each of which could be further subclustered into transcriptionally unique populations. From our initial fluorescence imaging studies, a role for intrinsic pacemaking by LMCs (*van Helden, 1993*; *von der Weid et al., 2008*), or by a novel population of CD34$^+$ lymphatic ICLCs (*McCloskey et al., 2002*; *Briggs Boedtkjer et al., 2013*), also referred to as telocytes, was further examined . We utilized PdgfrαCreER$^{TM}$ to further test whether these cells exhibited pacemaker capabilities. However, these PDGFRα$^+$ cells had minimal $Ca^{2+}$ activity despite ongoing contractions, and optogenetic stimulation of ChR2 in these cells failed to drive a spontaneous contraction. In contrast, photo-stimulation of LMCs expressing ChR2 elicited robust, propagated contractions with similar characteristics and propagation to spontaneous contractions in the same vessels. Furthermore, $Ca^{2+}$ imaging in LMCs revealed diastolic $Ca^{2+}$ transients in diastole that increased in frequency and spatial spread as pressure was elevated. We also demonstrated that the primary component of the AP driving the frequency change with pressure is diastolic depolarization, which we have previously reported to be dependent on ANO1 (*Zawieja et al., 2019*) and IP3R1 (*Zawieja et al., 2023*). Notably, we recently reported that diastolic $Ca^{2+}$ transients are abrogated in IALVs from Myh11CreER$^{T2}$-*Itpr1* inducible knockout mice, supporting an IP$_3$R1-ANO1 axis as the pressure-dependent pacemaker mechanism in LMCs. These results, in addition to the recent findings using targeted deletion of *Gjc1* (*Castorena-Gonzalez et al., 2018b*) or *Cacna1c* (*To et al., 2020*; *Davis et al., 2022*) in lymphatic muscle, support the model of LMCs as the intrinsic pacemaker, as has been previously proposed (*van Helden, 1993*; *Helden and Crowe, 1996*; *van Helden and Zhao, 2000*).

## Pacemaking in smooth muscle

In many smooth muscle organs, regulation of a coordinated contraction is a complex and multicellular phenomenon. Multiple cell types integrate physical and biological information into electrical activity to be transmitted to the force-producing smooth muscle cells, sometimes across great distances relative to cell size, to regulate $Ca^{2+}$ influx by voltage-dependent $Ca^{2+}$ channels required for contraction. The intestine is one such documented tissue in which cKIT$^+$ ICCs and PDGFRα$^+$ interstitial cells form an electrical syncytium to regulate intestinal motility (*Sanders et al., 1999*; *Sanders et al., 2014*). The pacemaking function of intestinal ICCs relies heavily on ANO1, a $Ca^{2+}$-activated Cl$^-$ channel, which is required for slow wave activity in the ICCs. Both cKIT and ANO1 can be used as a marker for ICCs in the intestine (*Hwang et al., 2009*; *Cobine et al., 2017*; *Malysz et al., 2017*), cKIT$^+$ and VIMENTIN$^+$ ICLCs have been observed in sheep lymphatic vessels (*McCloskey et al., 2002*), yet these cell populations did not form gap junctions with the smooth muscle to form electrical connections (*Briggs Boedtkjer et al., 2013*) as occurs in the intestinal ICCs. Our cKIT staining and CkitCreER$^{T2}$-*Rosa-26$^{mTmG}$* reporter studies on mouse IALVs revealed a sparse population of large ovoid cells previously classified as mast cells (*Chatterjee and Gashev, 2012*; *Zawieja et al., 2019*). Their identity as mast cells was further supported by a sustained global $Ca^{2+}$ event after stimulation with the mast cell degranulating agent compound 48–80. However, both VIMENTIN and CD34 showed robust staining throughout the mouse lymphatic vessel wall. LECs stained for VIMENTIN, as did non-muscle stellate-shaped cells, with many co-expressing CD34. Other smaller circular cells, some of which were also

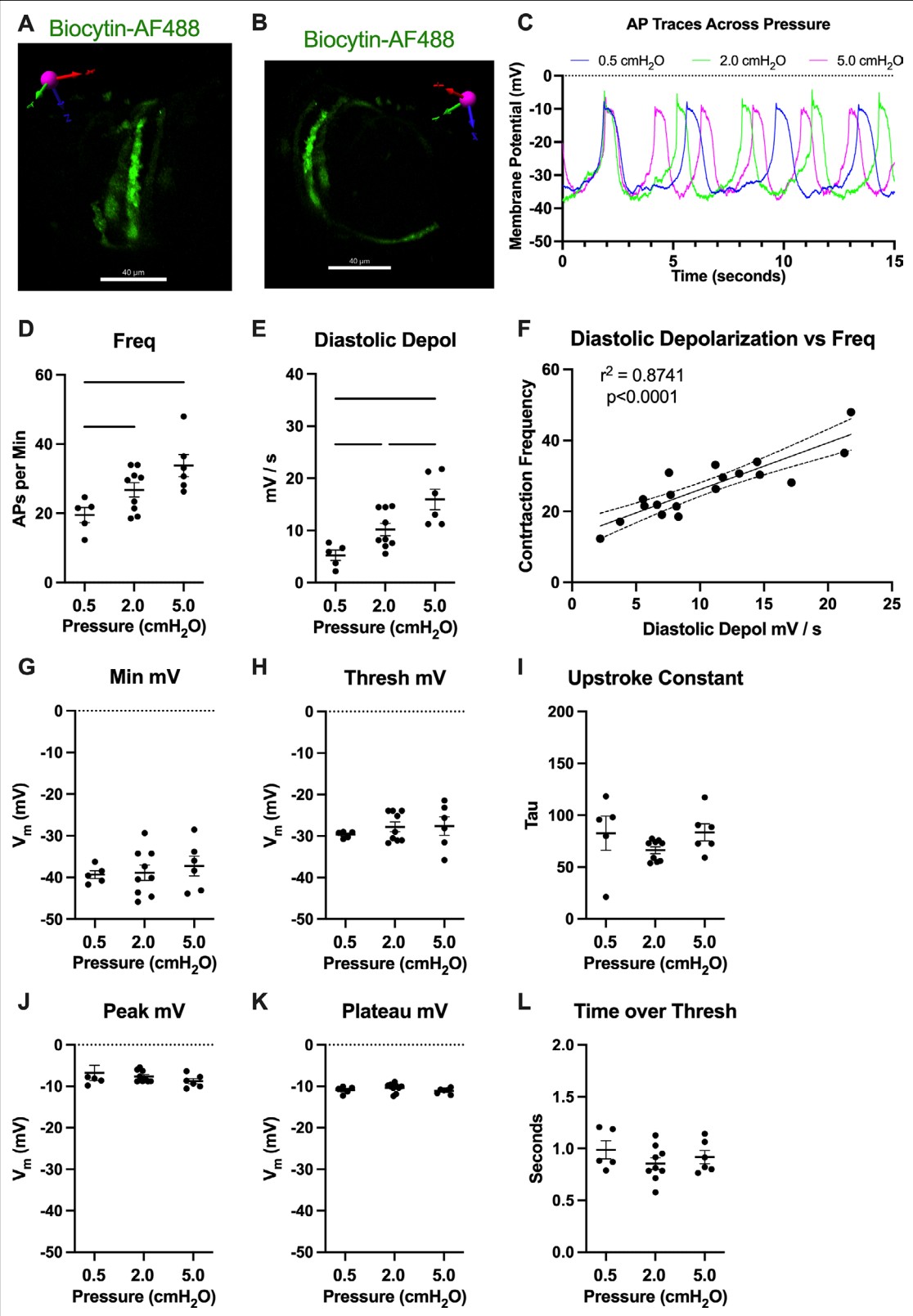

**Figure 13.** Pressure-dependent diastolic depolarization in lymphatic muscle cells (LMCs). Intracellular recordings of LMC action potentials (APs) were confirmed by loading (greater than 10 min) the impaling electrode with 1 M KCl 100 µg/ml AF488-Biocytin while recording APs followed by imaging on a spinning disk confocal microscope (*n* = 3 vessels from 3 mice). 3D reconstruction of the z-stack confirmed the circumferential pattern of the impaled LMC that was strongly labeled by AF488-Biocytin (**A**, **B**), which also labeled neighboring LMCs, likely through gap junctions as AF488-Biocytin is <1

*Figure 13 continued on next page*

*Figure 13 continued*

kDa. In a separate set of experiments APs were recorded at three different pressures, 0.5, 2, and 5 cmH$_2$O. We plotted the representative recordings from one cell at each pressure (**C**). AP frequency was significantly increased with pressure (**D**) as was the diastolic depolarization rate (**E**). Plotting the AP frequency and diastolic depolarization rate from all recordings at each pressure (**F**) highlights the significant effect diastolic depolarization rate has on the AP frequency. Minimum membrane potential (**G**), threshold membrane potential of AP initiation (**H**), upstroke constant (**I**), peak membrane potential (**J**), plateau membrane potential (**K**), and time over threshold (**L**) are also reported, although not significant. Recordings are from 10 IALVs from 10 mice.

cKIT$^+$ as well and some whose morphology was similar to that of the macrophage staining profile of the GFP$^+$ cells in IALVs from MacGreen mice, were also VIMENTIN$^+$, consistent with previous reports of macrophage staining in cLVs (*Bridenbaugh et al., 2013b*; *Chakraborty et al., 2015*; *Zawieja et al., 2016*). While VIMENTIN$^+$ cells had a perinuclear staining profile, CD34 demarcated the cell membrane and was useful for assessing the morphology of these cells. Of particular interest, the VIMENTIN$^+$CD34$^+$ cells densely populated the length of the mouse IALV, with a majority displaying a flattened stellate morphology characterized by a classic rounded oak-leaf appearance, although some displayed fine dendrite-like extensions. Contrasting with the previous findings in the human thoracic duct (*Briggs Boedtkjer et al., 2013*), we did not observe a significant population of CD34$^+$ cells with a bipolar morphology oriented axially along the vessel. However, z-stack reconstructions of sections of the mouse IALV that included the secondary valves revealed interstitial CD34$^+$PDGFR$\alpha^+$ cells that resembled those bipolar cells with multiple axon-like extensions throughout the endothelial leaflets; these were similar to interstitial cells that were previously reported in collecting vessel valves (*Leak and Burke, 1968*) and lymphovenous valves (*Geng et al., 2016*). While these cells have not been frequently described in the valves of peripheral cLVs, we observed them in each of the valve regions imaged and, in addition, they were labeled with other Cre drivers, including Cspg4Cre-*Rosa26$^{mTmG}$* and PdgfrβCreER$^{T2}$-*Rosa26$^{mTmG}$* (data not shown). Whether these cells regulate leaflet extracellular matrix deposition or lymphatic valve integrity is unknown, but a possible role as a critical pacemaker can be excluded as vessel segments without valves display normal contractile behavior (*van Helden, 1993*; *Gashev et al., 2002*). Instead, the majority of the CD34$^+$PDGFR$\alpha^+$ cells were found in the adventitia, in two to three layers, overtop the LMCs, and they were consistently observed in high density along the IALV. Some CD34$^+$PDGFR$\alpha^+$ cells or their extensions were present between the lymphatic endothelial and muscle layers, as had been previously reported with electron microscopy of human lymphatic vessels (*Briggs Boedtkjer et al., 2013*). Thus, while some of these AdvCs may be contained within the extracellular matrix that retracts onto the vessel during microdissection, many others are intimately dispersed within the vessel wall.

## PDGFRα$^+$CD34$^+$ cells are not involved in cLV pacemaking under physiological conditions

Co-staining of CD34 and PDGFRα has recently been ascribed as a delineating feature of telocytes, although PDGFRα routinely labels fibroblasts and specific interstitial cells in the GI tract involved in purinergic neurotransmission in the GI tract (*Kurahashi et al., 2011*; *Kurahashi et al., 2013*; *Clayton et al., 2022*). *Cd34* expression is also ascribed to some multipotent cell populations of various origins (*Sidney et al., 2014*). For example, PDGFRα$^+$ fibroblasts appear to be progenitors of the smooth muscle fibers associated with the lacteal, the lymphatic capillary in the villus (*Sanketi et al., 2024*). It remains controversial to what extent telocytes are distinct from or are components/subtypes of either cell type and morphological discrimination between the populations typically requires electron microscopy imaging (*Clayton et al., 2022*). Mesenchymal stromal cells (*Andrzejewska et al., 2019*) and fibroblasts (*Muhl et al., 2020*; *Buechler et al., 2021*; *Forte et al., 2022*) are not monolithic in their expression patterns displaying both organ-directed transcriptional patterns as well as intra-organ heterogeneity (*Lendahl et al., 2022*) as readily demonstrated by recent single-cell RNA sequencing studies that provided immense detail about the subtypes and activation spectrum within these cells and their plasticity (*Luo et al., 2022b*). We were able to distinguish up to 10 subclusters of AdvCs, the majority of which expressed or co-expressed *Cd34* and *Pdgfra*. These cells were consistently negative for smooth muscle markers such as *Des*, *Cnn1*, *Acta2*, *Myh11*, or the pericyte marker *Mcam*. However, *Pgfrb* expression was noted in our scRNAseq analysis and in our RT-PCR of sorted PdgfrαCreER$^{TM}$-*Rosa26$^{mTmG}$* cells. PDGFRβ protein expression was confirmed with variable immunofluorescence staining amongst the PDGFRα stained cells as well as LMCs. The PdgfrβCreER$^{T2}$*Rosa26$^{mTmG}$*

mice had only modest recombination in both the LMC and PDGFRα⁺ cell populations, but potentially highlighted a myofibroblast-like cell subpopulation, cells that might lie on the spectrum of differentiation from lymphatic muscle and PDGFRα⁺ cells, or perhaps a cell with pacemaker activity as PDGFRβ is widely used as a pericyte marker and some pericytes display pacemaker activity (*Hashitani et al., 2015*). Adding to this intrigue, the PdgfrαCreER™ sorted cells expressed transcripts for *Cacna1c*, the voltage-gated L-type $Ca^{2+}$ channel critical for lymphatic contractions (*Zawieja et al., 2018a*; *To et al., 2020*); *Ano1*, the ion channel underlying pressure-dependent chronotropy (*Mohanakumar et al., 2018*; *Zawieja et al., 2019*); and *Gjc1*, the primary connexin mediating electrical conduction in mouse lymphatic collecting vessels (*Castorena-Gonzalez et al., 2018b*; *Hald et al., 2018*). Expression of these genes in certain subpopulations of the AdvCs was also apparent in our scRNAseq analysis. Thus, the presence of those gene transcripts does not appear to be due to muscle cell contamination or incidental recombination in LMCs, as we did not detect LMC markers in the RT-PCR profiling of the sorted PDGFRα⁺ cells, nor were GFP-expressing cells with an LMC morphology observed in imaging of PdgfrαCreER™-*Rosa26^{mTmG}* vessels. Critically, however, deletion of *Cacna1c*, *Gjc1*, or *Ano1* through PdgfrαCreER™-mediated recombination neither recapitulated the previous phenotypes achieved with Myh11CreER^{T2} (*Castorena-Gonzalez et al., 2018b*; *Zawieja et al., 2019*; *To et al., 2020*; *Davis et al., 2022*) nor significantly affected pacemaking in mouse popliteal cLVs. This finding is in stark contrast to the complete lack of contractions observed in Myh11CreER^{T2}-*Cacna1c*^{fl/fl} vessels (*To et al., 2020*; *Davis et al., 2023b*) or the vessels from vascular muscle specific *Itga8CreER*^{T2}-*Cacna1c*^{fl/fl} mice (*Davis et al., 2022*; *Warthi et al., 2023*), and the significant loss in pressure-induced chronotropic modulation of pacemaker function in IALVs with Myh11CreER^{T2}-mediated deletion of *Ano1* that we have previously reported (*Zawieja et al., 2019*). While a subpopulation of CD34⁺PDGFRα⁺ cells may share expression of critical pacemaker genes identified in the LMCs, they do not appear to be involved in cLV pacemaking or contractile function under physiological states. Instead, CD34⁺PDGFRα⁺ cells co-stained significantly with LY6A⁺, suggesting they may be primed to act as resident multipotent cells (*Song et al., 2020*; *Kimura et al., 2021*). To this point, the PdgfrαCreER™ FACS-purified cells also expressed markers associated with 'stemness'such as *Cd34*, *Klf4*, *Gli1*, *Itgb1*, *Eng*, *Cd44*, and *Vimentin*, in addition to *Ly6a*, and it is likely that the PdgfrαCreER™ population includes various distinct subpopulations (*Jolly et al., 2022*) expressing these markers. These cells may play a role in rebuilding the lymphatic collecting vessel vasculature following collecting vessel damage or lymph node resection, and further studies are required to assess their functional multipotency.

## SR $Ca^{2+}$ cycling in pacemaking

The use of the mouse IALV model, in addition to the simplicity of the vessel architecture, provided the use of genetic tools that previously had been instrumental in identifying the cKIT⁺ ICC as the pacemaker cells of the GI tract (*Ward et al., 1994*; *Huizinga et al., 1995*; *Torihashi et al., 1995*). Through the use of the respective PdgfrαCreER™ and Myh11CreER^{T2} drivers, we were able to specifically image $Ca^{2+}$ in each respective cell type in pressurized, contracting vessels. Pacemaking initiating cells have an inherently unstable membrane potential, oftentimes utilizing the oscillatory nature of $Ca^{2+}$ release from the sarcoendoplasmic reticulum coupled to $Ca^{2+}$-sensitive electrogenic exchangers or ion channels to drive depolarization (*van Helden, 1993*; *Hashitani et al., 2015*; *Baker et al., 2021b*; *Sanders et al., 2022*). One such example is the pacemaker ICC in the gastric corpus which exhibits abundant $Ca^{2+}$ transients that couple to ANO1-mediated chloride currents in both the intervening period between slow waves as well as the plateau phase of the slow wave (*Baker et al., 2021a*); however, such activity is not characteristic of all pacemaker ICC types. The identification of a $Ca^{2+}$-activated chloride current in LMCs (*van Helden, 1993*; *Toland et al., 2000*) and its correspondence with subcellular $Ca^{2+}$ transients (*van Helden, 1993*; *Ferrusi et al., 2004*; *von der Weid et al., 2008*) led Van Helden to postulate that LMCs had intrinsic pacemaking capability (*van Helden, 1993*; *Helden and Crowe, 1996*). We have previously reported that mouse LMCs in pressurized vessels routinely display subcellular $Ca^{2+}$ release events that reflect the kinetics and characteristics of $Ca^{2+}$ puffs and waves in addition to the coordinated global $Ca^{2+}$ flash associated with $Ca^{2+}$ influx during an AP (*Castorena-Gonzalez et al., 2018b*; *Zawieja et al., 2018a*; *Zawieja et al., 2019*). Here we confirmed the consistent presence of subcellular $Ca^{2+}$ transients only in LMCs with GCaMP6f driven by Myh11CreER^{T2} but not in the cells with GCaMP6f driven by PdgfrαCreER™. Critically, we also demonstrated that the $Ca^{2+}$ transients increased in both frequency and spatial spread as pressure was elevated in the vessel,

as would be expected to account for the pressure-dependent chronotropy observed in lymphatic collecting vessels. This underscores the finding that genetic deletion of *Ano1* in the LMCs dramatically reduced contraction frequency and abolished pressure-dependent chronotropy in those vessels (*Zawieja et al., 2019*). This phenotype was largely replicated with a similar reduction in frequency and loss of pressure-dependent chronotropy in our recent study utilizing Myh11CreER$^{T2}$ to drive deletion of IP3R1 from LMCs (*Zawieja et al., 2023*) in which these diastolic Ca$^{2+}$ transients were absent. This fits with the central role of IP3R and subcellular Ca$^{2+}$ release as critical components of intrinsic LMC pacemaking (*Helden and Crowe, 1996*; *von der Weid et al., 2008*). In addition to the transcriptional heterogeneity identified by scRNASeq, we also noted heterogeneity in the propensity of LMCs to display diastolic Ca$^{2+}$ transients under control conditions or show the sustained Ca$^{2+}$ oscillations that occur in the presence of nifedipine. We did not detect any significant difference in the expression of *Itpr1*, the gene encoding the IP3R1, across our LMCs subclusters. However, when using less stringent conditions, we identified that the LMC cluster "0" had significantly increased expression of *Itprid2* (Log2FC of 0.26), which encodes the Kras-induced actin-interacting protein (KRAP). KRAP has recently been implicated in IP3R1 immobilization and licensing and was required for IP3R1-mediated Ca$^{2+}$ puffs (*Thillaiappan et al., 2021*; *Atakpa-Adaji et al., 2024*). Whether the higher expression of KRAP results in a greater probability of these LMCs to display IP3R1-dependent Ca$^{2+}$ oscillations in LMCs requires further investigation. Of note, LMCs also express the components for store-operated calcium entry including *Stim1*, *Stim2*, *Orai1*, *Orai3*, *Saraf*, and *Stimate*, which may be involved in maintaining IP3R1-dependent SR Ca$^{2+}$ release oscillations.

The membrane potential recordings made in this study suggest that the regulation of pressure-dependent chronotropy is through modulation of the diastolic depolarization rate in LMCs, as previously demonstrated in rat mesenteric lymphatic vessels (*Zawieja et al., 2018b*). The appearance of the diastolic depolarization may depend on the method of vessel stretch employed as it is not always observed in wire myograph preparations (*von der Weid et al., 2014*). Notably, in this study, PdgfrαCreER$^{TM}$-mediated deletion of Ano1 had no effect on contractile parameters. The lack of Ca$^{2+}$ transients in PDGFRα$^+$ cells across any stage of the lymphatic contraction cycle also diminishes any expected role for this cell type to perform as the pacemaker for the mouse IALV. We recently showed that pressure-dependent Ca$^{2+}$ mobilization from the SR, through IP3R1 (*Zawieja et al., 2023*), sets the basis for LMC pacemaking as previously proposed (*van Helden, 1991*; *von der Weid et al., 2008*). However, the mechanisms driving IP3R1 activation and Ca$^{2+}$ oscillations remain to be fully addressed.

A pacemaker cell would be expected to be electrically coupled to the LMC layer to permit the nearly synchronous conduction velocity of the contraction wave (*Zawieja et al., 1993*; *Castorena-Gonzalez et al., 2018b*; *Hald et al., 2018*) and to transmit depolarization into coupled LMCs to activate the voltage-dependent Ca$^{2+}$ channels that are responsible for lymphatic muscle APs. Connexins are the molecular constituents of gap junctions and, as stated above, we detected *Gjc1* expression in PdgfrαCreER$^{TM}$ sorted cells. However, we did not detect any impairment in pacemaking, nor were contraction conduction speed deficits or multiple pacemakers noted in the PdgfrαCreER$^{TM}$ -*Gjc1*$^{fl/fl}$ popliteal cLVs, in contrast to the development of multiple pacemaker sites and the lack of entrainment that characterize cLVs from Myh11CreER$^{T2}$-*Gjc1*$^{fl/fl}$ mice (*Castorena-Gonzalez et al., 2018b*). Admittedly, we did not perform an exhaustive assessment of the connexin expression profile of the CD34$^+$PDGFRα$^+$ cells, and *Gjc1* may not be the dominant connexin expressed in the CD34$^+$PDGFRα$^+$ cells, or heterotypic connexons could exist (*Koval et al., 2014*). However, electron microscopy studies of the putative ICLC in the human thoracic duct did not detect any gap junctions, although peg-and-socket connections were observed (*Briggs Boedtkjer et al., 2013*). We utilized optogenetics to directly depolarize the specific cell populations in both the PdgfrαCreER$^{TM}$ and Myh11CreER$^{T2}$ mouse models in an attempt to drive contractions. Local photo-stimulation of the PDGFRα cells failed to initiate contraction, while the stimulation of Myh11CreER$^{T2}$ recombined cells resulted in contractions that were indistinguishable from the spontaneously occurring contractions. These results give functional credence to the lack of hetero-cellular coupling via gap junctions that was previously reported (*Briggs Boedtkjer et al., 2013*). Just as critically, our results also highlight the regenerative nature of the lymphatic muscle AP. Local, optogenetic-initiated depolarization of either a single or a few LMCs to threshold was sufficient to drive a coordinated contraction along the vessel demonstrating conducted activity at the tissue level.

## Conclusions

Our present findings lend further support to the hypothesis that the LMCs are intrinsic pacemakers (*van Helden et al., 2006*; *Mitsui and Hashitani, 2020*) and that mouse cLVs do not require an ICC-like cell network to drive propagated contractions. These findings also underscore the significance of lymphatic muscle $Ca^{2+}$ handling as the driver of lymphatic pacemaking, which can be compromised in disease states leading to impaired lymphatic contractile activity (*Stolarz et al., 2019*; *Lee et al., 2020*; *Van et al., 2021*). Further studies delineating the specific SR $Ca^{2+}$ release and influx pathways, and the contributions of $Ca^{2+}$ sensitive ion channels are required to develop sophisticated in silico models and identify potential therapeutic targets to rescue lymphatic pacemaking in lymphedema patients (*Olszewski, 2002*; *Olszewski, 2008*).

## Limitations

One fundamental assumption underlying our conclusions is that there is a conserved pacemaking pathway and cell type regulating lymphatic collecting vessel contractions across species, specifically pertaining to the capability of lymphatic muscle to maintain pacemaking and coordination despite changes in tissue complexity and cLV wall thickness. It is worth noting that lymphatic collecting vessels in mice have similar pressure-dependent chronotropy and contraction conduction velocity as recorded in rats and human vessels (*Castorena-Gonzalez et al., 2018b*). These similarities exist despite the fact that mouse lymphatic collecting vessels are typically encircled by a single layer of lymphatic muscle, while larger species may have multiple layers of LMCs in the wall. It is possible that vessels with multiple layers of LMCs need a network of ICLC to coordinate their activity. The simplicity in the makeup of the mouse cLV and the use of cell targeting Cre models provides great control over experimental variables, but other cell types may be required for coordination of LMC pacemaking in other species where the lymphatic cLV walls are larger and thicker and contain multiple muscle cell layers. Our scRNAseq analysis also is likely biased using *Rosa26*<sup>mTmG</sup> mice with FACS purification to remove debris and concentrate specific cell types from these pooled small vessels. Larger and more complex cells, with attributes that can be ascribed to ICCs, are more likely to be lost in this methodology (e.g., depending on the FACS gating parameters), and this procedure can also elicit a stress response in the transcriptome of the analyzed cells. However, we also did not observe long and complex cells, aside from the circumferential LMCs, in our immunofluorescence and recombination reporter imaging experiments. Immediate and early gene expression motifs driven by a stress response may be a component of the differences in subclusters that were identified. Future scRNAseq or snRNAseq studies with deeper sequencing will be required to ensure that the full transcriptomic heterogeneity is accounted for under different cellular stress conditions.

Our data demonstrate that limited staining of a few cell markers alone is insufficient to identify discrete cell populations in mouse cLVs. Additionally, mRNA expression does not equal protein translation nor guarantee specific function as we did not readily detect endothelial CD34 with immunofluorescence despite detecting transcript; additionally, PdgfrαCreER<sup>TM</sup>-mediated deletion of *Ano1*, *Gjc1*, or *Cacna1c* had no effect on cLV pacemaking. Further experimentation is also required to fully characterize expression of multipotent cell markers and function of CD34<sup>+</sup>PDGFRα<sup>+</sup>LY6A<sup>+</sup> cells invested within the mouse cLVs, although doing so was beyond the scope of this study assessing pacemaker identity. Tangentially, another limitation of our approach pertains to the specificity and recombination efficiency of inducible Cre recombinase models, which can be a notable confounding variable (*Chakraborty et al., 2019*). We observed that our inducible Cre models led to a degree of nonspecific recombination within the mouse cLV, with GCaMP6f and ChR2 particularly susceptible to recombination compared to the *Rosa26*<sup>mTmG</sup> reporter. Recombination in multiple cell types was expected with the constitutive Cre models we employed (Cspg4Cre and PdgfrαCre), as vascular and lymphatic muscle precursor cells can transiently express *Nestin*, *Pdgfra*, and *Cspg4* (*Hill et al., 2015*; *Castorena-Gonzalez et al., 2018b*; *Kenney et al., 2020*). We also observed that PdgfrβCreER<sup>T2</sup> drove recombination in a subpopulation of LMCs and PDGFRα<sup>+</sup> cells. These appeared to be two distinct populations that only share expression for *Pdgfrb* based on our scRNAseq dataset, but which may exist along a continuum of differentiation. PDGFB–PDGFRβ signaling is critical for normal mural cell recruitment to both the blood and lymphatic vasculature (*Gaengel et al., 2009*; *Wang et al., 2017*) and proliferating vascular smooth muscle cells and pericytes have both been documented to express *Pdgfrb* (*Andrae et al., 2008*; *Pitulescu and Adams, 2014*). Ideally, novel Cre or combinatorial Cre models

that specifically target LMCs or subpopulations of LMCs may be developed to further tease out the functional roles of these cells.

## Materials and methods

### Mice

Wild-type male mice (25–35 g) on the C57BL/6J background, *Rosa26^{mTmG}* reporter (*Muzumdar et al., 2007*) (Strain#007676), transgenic PdgfrαCre (Strain#013148), CSFR1-EGFP (MacGreen) (*Sasmono et al., 2003*) (Strain#018549), genetically encoded Ca²⁺ sensor GCaMP6f (*Chen et al., 2013*) (Strain#028865), transgenic PdgfrαCreER™ (*Kang et al., 2010*) (Strain#018280), Cspg4-Cre (Strain #:008533) (*Zhu et al., 2008*), and ChR2/tdTomato fusion mice (*Madisen et al., 2012*) (Strain#012567) were purchased from The Jackson Laboratory (Bar Harbor, MA, USA). PdgfrβCreER^{T2} (*Gerl et al., 2015*) mice were a gift from Ralf Adams (Mac Planck Institute) and kindly provided by Lorin Olson (Oklahoma Medical Research Foundation) and are currently available at Jax (Strain#029684). The Myh11CreER^{T2} mice (*Wirth et al., 2008*) were a gift from Stefan Offermanns, Max-Planck-Intstitut für Herz- und Lungenforschung, Bad Nauheim, Germany, and are currently available at Jax (Strain #019079, Y-Linked). CkitCreER^{T2} mice (*Heger et al., 2014*) were a gift from Dieter Saur (Technical University of Munich). Prox1-eGFP mice (*Choi et al., 2011*) were a gift from Young-Kwon Hong (University of Southern California). For genotyping, we isolated genomic DNA from mouse tail clips using the HotSHOT method (*Truett et al., 2000*). Specific mouse genotypes were confirmed via PCR using 2x PCR Super Master Polymerase Mix (Catalog # B46019, Bimake, Houston, TX) performed as specified by the provider. Mice used for this study were 3–12 months of age. All animal protocols were approved by the University of Missouri Animal Care and Use Committee (Protocol 53461 and 41500) and conformed to the US Public Health Service policy for the humane care and use of laboratory animals (PHS Policy, 1996).

### Inducible Cre tamoxifen induction

Mice harboring PdgfrαCreER™, PdgfrβCreER^{T2}, Myh11CreER^{T2}, and CkitCreER^{T2} were crossed with *Rosa26^{mTmG}* mice to generate PdgfrαCreER™-*Rosa26^{mTmG}*, PdgfrβCreER^{T2}-*Rosa26^{mTmG}*, Myh11CreER^{T2}-*Rosa26^{mTmG}*, and CkitCreER^{T2}-*Rosa26^{mTmG}* mice, respectively. The resulting inducible Cre-*Rosa26^{mTmG}* mice were induced with tamoxifen 2–4 weeks after weaning for confocal imaging. Mice aged 2–6 months were injected with tamoxifen for contraction studies, FACS analysis, GCaMP6f imaging, and Chr2 induction. Tamoxifen induction was performed via consecutive 100 µl i.p. injections of tamoxifen ranging from 1 to 5 days at concentrations ranging from 0.2 to 10 mg/ml in safflower oil, using a titrated induction protocol to determine the extent of recombination in specific cell populations. We used our maximal induction protocol, 100 µl of tamoxifen at 10 mg/ml over 5 consecutive days, for CkitCreER^{T2}-GCaMP6f, Myh11CreER^{T2}-GCaMP6f, and PdgfrαCreER™-GCaMP6f mice. Due to the paucity of recombined cells in the CkitCreER^{T2}-*Rosa26^{mTmG}* reporter mice, we used our maximal tamoxifen induction protocol for CkitCreER^{T2}-ChR2/tdTomato mice as this still resulted in the ability to excite single recombined cells. Myh11CreER^{T2}-ChR2/tdTomato mice were induced with one 100 µl i.p. injection of tamoxifen at 0.2 mg/ml while PdgfrαCreER™-ChR2/tdTomato mice were induced with 1 injection at 0.4 mg/ml tamoxifen to get mosaic induction sufficient for single-cell stimulation. All mice, regardless of induction duration, were given at least 2 weeks to recover following tamoxifen injection.

### Lymphatic vessel isolation

We utilized both popliteal and inguinal-axillary lymphatic collecting vessels (IALVs) in this study, which were isolated as described previously (*Zawieja et al., 2018a*). In brief, mice were anesthetized with a cocktail of 100/10 mg/ml ketamine/xylazine and shaved along the flank or the legs for IALVs and popliteal cLVs, respectively. The IALV (also referred to as the flank cLV) is located adjacent to the thoracoepigastric vein and connects the inguinal and axillary lymph nodes. A cut was made along the dorsal midline, and the skin was retracted and pinned out to reveal the thoracoepigastric vascular bed. The thoracoepigastric vascular bed and connected perivascular adipose containing the IALV vessels was dissected out and pinned onto a Sylgard-coated dish in Krebs buffer. Popliteal lymphatic vessels were exposed through a superficial incision in the leg, removed, and transferred to the Krebs-albumin filled dissection chamber. After removal, the vessel was carefully cleaned of adipocytes and excess

matrix using fine forceps and scissors through micro-dissection. For immunofluorescence, sections containing two to three valves were isolated, while shorter IALV sections consisting of one to two valves were isolated for GCaMP6f $Ca^{2+}$ imaging. Similarly, popliteal cLVs were isolated (*Castorena-Gonzalez et al., 2018a*) following an incision along the skin overlying the saphenous, removed, and transferred to the Krebs-albumin filled dissection chamber; these vessels were used for ChR2 optogenetic depolarization experiments.

## Lymphatic vessel isobaric function

PdgfrαCreER™ mice were crossed with *Ano1*$^{fl/fl}$, *Gjc1*$^{fl/fl}$, and *Cacna1c*$^{fl/fl}$ mice to generate PdgfrαCreER™-*Ano1*$^{fl/fl}$, PdgfrαCreER™-*Gjc1*$^{fl/fl}$, and PdgfrαCreER™-*Cacna1c*$^{fl/fl}$ mice. These mice and their respective 'fl/fl' controls were injected with tamoxifen as described above for 5 days and given 2 weeks to recover. The popliteal vessels were isolated, cleaned, and prepared for isobaric contractile tests as previously reported (*Davis et al., 2023a*). Once equilibrated, inner diameter was tracked over a physiological pressure range (stepped from 3 to 2, 1, 0.5, 3, 5, 8, and 10 cmH$_2$O) with 2 min of recording at each pressure. Following the pressure step protocol, the vessels were equilibrated in $Ca^{2+}$-free Krebs buffer (3 mM EGTA), and diameter at each pressure recorded under passive conditions (DMAX). The contractile parameters end diastolic diameter (EDD), end systolic diameter (ESD), and contraction frequency (FREQ) were assessed from the diameter trace using LabVIEW:

1. Contraction amplitude (AMP) = EDD − ESD
2. Normalized contraction amplitude = ((EDD − ESD)/DMAX) × 100
3. Ejection fraction (EF) = (EDD$^2$ − ESD$^2$)/EDD$^2$
4. Fractional pump flow (FPF) = EF × FREQ
5. Tone = ((DMAX − EDD)/DMAX) × 100

## Methylene blue staining

Isolated IALVs sections were transferred into a Krebs-BSA buffer filled 3 ml observation chamber, with a cover slip bottom, and cannulated onto two glass micropipettes (30–80 µm, outer diameter) held in place by pipette holders on a Burg-style V-track mounting system. The pipette holders were attached to a three-way valve stopcock with polyethylene tubing filled with Krebs-BSA buffer. Vessels were pressurized to approximately 5 cmH$_2$O by raising the three-way valve and the vessels were stretched to remove any slack. For methylene blue (Sigma, M9140) staining, IALVs from wild-type C57Bl6 mice were stained with 50 µM methylene blue in Krebs-BSA buffer for 2 hr at room temperature and covered in foil to limit light-induced phototoxicity. After the staining period, the vessel chambers were washed three times with $Ca^{2+}$-free PSS to remove methylene blue. Brightfield images and manual Z-stack videos were collected on an inverted Leica DMi1 ×4 or ×20 air objective, or a Leica DMi8 with a ×25 water objective or an inverted DMi8 using a Leica Flexacam C1 color camera for image acquisition. Some methylene blue images were also collected using a color Nikon DS-Fi3 camera. The collected z-stacks were analyzed using ImageJ and the 'Stack Focuser' plugin (https://imagej.nih.gov/ij/plugins/stack-focuser.html). To accentuate the methylene blue-stained cells, the color image stack was split into red, green, and blue channel stacks. The blue channel stack was then divided by the green channel stack using the 'Image Calculator' function. The resulting 32-bit image was then converted into a 16-bit image to permit the use of the Stack Focuser plugin with the 'n kernel value' set to 11.

## Fluorescence confocal imaging

IALV vessels from each respective inducible Cre-*Rosa26*$^{mTmG}$ mouse were prepared in a similar manner (excluding the addition of methylene blue). We performed confocal imaging to acquire z-stacks of 7–10 overlapping regions of interest to allow for manual stitching, with 1 µm z-steps at (20X) or 0.5 µm steps at 40X. We imaged through to the midpoint of the vessel except when imaging the valve interstitial cells, in which case the vessel lumen was imaged. Max projections were made using FIJI. Following live imaging, the vessels were pressurized to 5 cmH$_2$O and fixed with 4% paraformaldehyde for 30 min at room temperature. IALVs were then washed with PBS containing 0.1% Triton X-100 (PBST) 3 times and blocked for a minimum of 2 hr with Blockaid (B-10710, Thermo Fisher Scientific). IALVs were then stained with the corresponding primary antibodies in BlockAid Solution: anti-smooth muscle actin (ACTA2) 1:500 (Sigma, A2547), anti-GFP 1:200 (Thermo Fisher,

A11122), anti-cKIT 1:100 (Cell Signaling, 3074), anti-VIMENTIN 1:100 (Thermo Fisher, OMA1-06001), anti-desmin 1:200 (Invitrogen, PA5-16705), anti-GFP 1:200 (Abcam, ab13970), anti-CD34 1:200 (Invitrogen, 14-0341-82), anti-PDGFR$\mathrm{A}$ 1:200 (R&D Systems, AF1062), anti-PDGFRβ 1:200 (eBiosciences, 14-1402-82), anti-calponin 1:500 (Abcam, AB46794), anti-MYH11 1:500 (Abcam, AB124679), anti-LY6A 1:200 (Biolegend, 108101). IALVs were then washed in PBS and incubated overnight with the corresponding donkey secondary antibodies (Thermo Fisher) at 1:200. After a final wash, IALVs were re-cannulated and pressurized for imaging using the spinning disk confocal microscope and Hamamatsu Orca Flash4 camera using a 20X air objective (Olympus UplanApo, 0.75) or 40X (Olympus UApo A340, 1.15) water objective. Images were taken as described above, and the resulting stacks were turned into a max projection using FIJI. Colocalization analysis of the max projections of CD34 and PDGFRα was performed using the BIOP JACoP colocalization plugin (*Bolte and Cordelières, 2006*) with both Pearson's and Mander's coefficients reported.

## LMC dissociation and FACS collection

IALVs vessels PdgfrαCreER$^{TM}$-*Rosa26$^{mTmG}$*, PdgfrβCreER$^{T2}$-*Rosa26$^{mTmG}$*, Myh11CreER$^{T2}$-*Rosa26$^{mTmG}$*, Macgreen, and Prox1-eGFP mice were dissected and cleaned of excess adventitia and adipose tissue in Krebs buffer. Isolated vessels were then transferred into a low Ca$^{2+}$ PSS solution supplemented with 0.1 mg/ml bovine serum albumin (BSA, Amersham Life Science, Arlington Heights, IL). Primary LMCs were collected by enzymatic dissociation of IALVs. The dissected vessels were cleaned in room temperature Krebs-BSA buffer and then transferred into a 1-ml tube of low-Ca$^{2+}$ PSS on ice, washed, and equilibrated for 10 min. Vessels were then digested in low-Ca$^{2+}$ PSS with 26 U/ml papain (Sigma, P4762) and 1 mg/ml dithioerythritol for 30 min at 37°C with gentle agitation every few minutes. This solution was then decanted and replaced with low-Ca$^{2+}$ PSS containing 1.95 collagenase H (U/ml, Sigma), 1.8 mg/ml collagenase F (Sigma), and 1 mg/ml elastase (Worthington LS00635) and incubated for 3–5 min at 37°C. The mixture was then spun down at 1000 rpm for 4 min, the digestion buffer removed, and replaced with low-Ca$^{2+}$ PS. This process was repeated twice to remove residual digestion buffer. The vessel was then triturated with a fire-polished Pasteur pipette to dissociate the cells into a single-cell suspension, passed through a Falcon cap strainer (35 μm), and resuspended in ice-cold low-Ca$^{2+}$ PSS for sorting. For inducible Cre-*Rosa26$^{mTmG}$* mice, GFP$^+$RFP$^-$ cells or GFP$^+$ cells from Macgreen and Prox1-eGFP mice were then FACS-purified straight into RNA isolation buffer for RT-PCR analysis. FACS was performed with a Beckman-Coulter MoFlo XDP instrument using an excitation laser (488 nm) and emission filter (530/40 nm). Sorting was performed using 70 μm nozzle at a sheath pressure of 45 psi and sort rate of 100 events/s and with an efficiency of >90%. To maximize cell yield, we isolated both the left and right full-length IALV vessels from two mice for digestions and subsequent FACS collection. For Myh11CreER$^{T2}$-*Rosa26$^{mTmG}$* and PdgfrαCreER$^{TM}$-*Rosa26$^{mTmG}$*, the yield averaged 1000–2000 cells per mouse. For Prox1-eGFP mice, LEC yield was typically 1500–2000 cells per mouse.

## RT-PCR profiling of FACS-purified cells

Total RNA was extracted from FACS-purified GFP$^+$ cells from the isolated IALVs vessels using the Arcturus PicoPure RNA isolation kit (Thermo Fisher Scientific, Waltham, MA) per the listed instructions. Prior to elution in 20 μl of water, on-column DNase digestion (QIAGEN, Valencia, CA) was performed to ensure removal of genomic DNA contaminants. RNA was converted into cDNA using SuperScript III First-Strand Synthesis System (Thermo Fisher Scientific, Waltham, MA) using oligo (dT) and random hexamer priming following the manufacturer's protocol. Each RT reaction used approximately 50–100 cells worth of RNA based on the sorted cells count number. Our PCR reaction mixture contained first-strand cDNA as the template, 2 mM MgCl$_2$, 0.25 μM primers, 0.2 mM deoxynucleotide triphosphates, and GoTaq Flexi DNA polymerase (Promega, Madison, WI). The PCR program comprised an initial denaturation step at 95°C for 4 min; followed by 35 repetitions of the following cycle: denaturation (94°C, 30 s), annealing (58°C, 30 s), and extension (72°C, 30 s). This was followed by a final elongation step for 5 min at 72°C. PCR amplification products were separated on a 2% agarose gel by electrophoresis, stained with SYBR-Safe (Thermo Fisher Scientific, Waltham, MA), and visualized by UV trans-illumination. All primers were designed to amplify an intron-spanning region. Endpoint RT-PCR Primer sequences, amplicon size, accession numbers, and source are listed in *Table 1*.

**Table 1.** Primer list for RT-PCR.

| Gene | Strand | Accession # | Sequence (5'–3') | Size | Exon | Source |
|---|---|---|---|---|---|---|
| Prox1 | s | NM_008937 | GTA AGA CAT CAC CGC GTG C | 218 | 1 | NIH Primer Tool |
| | as | | TCA TGG TCA GGC ATC ACT GG | | 2 | |
| Itgam (Cd11b) | s | NM_008401 | ATG GAC GCT GAT GGC AAT ACC | 203 | 13 | MGH Primer Bank ID 668048a1 |
| | as | | TCC CCA TTC ACG TCT CCC A | | 14 | |
| Pdgfra | s | NM_011058 | AGA GTT ACA CGT TTG AGC TGT C | 252 | 8 | MGH Primer Bank 26349287a1 |
| | as | | GTC CCT CCA CGG TAC TCC T | | 10 | |
| Myh11 | s | NM_013607 | AAG CTG CGG CTA GAG GTC A | 238 | 33 | MGH Primer Bank ID 7305295a1 |
| | as | | CCC TCC CTT TGA TGG CTG AG | | 34 | |
| cKit (Cd117) | s | NM_021099 | CGC CTG CCG AAA TGT ATG ACG | 162 | 21 | *Drumm et al., 2018* |
| | as | | GGT TCT CTG GGT TGG GGT TGC | | 23 | |
| Pdgfrb | s | NM_008809 | AGC TAC ATG GCC CCT TAT GA | 367 | 16 | *Basciani et al., 2004* |
| | as | | GGA TCC CAA AAG ACC AGA CA | | 19 | |
| Cdh5 (Cadherin, VE-cadherin) | s | NM_009868 | CTT CCT TAC TGC CCT CAT TGT | 313 | 3 | IDT Primer Quest |
| | as | | CTG TTT CTC TCG GTC CAA GTT | | 5 | |
| Nos3 (eNOS) | s | NM_008713 | CTG CCA CCT GAT CCT AAC TTG | 143 | 22 | IDT Real time primer tool |
| | as | | CAG CCA AAC ACC AAA GTC ATG | | 23 | |
| Acta2 (Smooth Muscle Actin) | s | NM_007392 | GAG CTA CGA ACT GCC TGA C | 129 | 7 | IDT TaqMan Mm.PT.58.16320644 |
| | as | | CTG TTA TAG GTG GTT TCG TGG A | | 8 | |
| Cacna1c exon1b | s | NM_001159533 | ATG GTC AAT GAA AAC ACG AGG ATG | | 1 | *Cheng et al., 2007* |
| | as | | GGA ACT GAC GGT AGA GAT GGT TGC | 234 | 2 | |
| Cd34 | as | NM_133654 | GGT ACA GGA GAA TGC AGG TC | 119 | 1 | IDT Mm.PT.58.8626728 |
| | s | | CGT GGT AGC AGA AGT CAA GT | | 2 | |
| Cspg4 (Ng2) | as | NM_139001 | CTT CAC GAT CAC CAT CCT TCC | 132 | 5 | IDT Mm.PT.58.29461721 |
| | s | | CCC GAA TCA TTG TCT GTT CCC | | 6 | |
| Vimentin | s | NM_011701 | CTG TAC GAG GAG GAG ATG CG | 249 | 1 | *Li et al., 2016* |
| | as | | AAT TTC TTC CTG CAA GGA TT | | 3 | |
| Desmin | s | NM_010043 | GTG GAT GCA GCC ACT CTA GC | 218 | 3 | MGH Primer Bank ID 33563250a1 |
| | as | | TTA GCC GCG ATG GTC TCA TAC | | 4 | |
| Mcam (Cd146) | s | NM_023061 | CCC AAA CTG GTG TGC GTC TT | 220 | 1 | MGH Primer Bank 10566955a1 |
| | as | | GGA AAA TCA GTA TCT GCC TCT CC | | 3 | |
| Klf4 | s | NM_010637 | ATT AAT GAG GCA GCC ACC TG | 400 | 1 | *Majesky et al., 2017* |
| | as | | GGA AGA CGA GGA TGA AGC TG | | 3 | |
| Ly6a (Sca1) | s | NM_001271416 | CTC TGA GGA TGG ACA CTT CT | 400 | 2 | *Majesky et al., 2017* |
| | as | | GGT CTG CAG GAG GAC TGA GC | | 4 | |
| Gli1 | s | NM_01029 | ATC ACC TGT TGG GGA TGC TGG AT | 316 | 8 | *Kramann et al., 2015* |
| | as | | CGT GAA TAG GAC TTC CGA CAG | | 10 | |
| Itgb1 (Cd29) | s | NM_010578 | TCG ATC CTG TGA CCC ATT GC | 170 | 14 | NIH Primer Tool |
| | as | | AAC AAT TCC AGC AAC CAC GC | | 15 | |

*Table 1 continued on next page*

*Table 1 continued*

| Gene | Strand | Accession # | Sequence (5'–3') | Size | Exon | Source |
|------|--------|-------------|------------------|------|------|--------|
| *Endoglin* (*Eng, Cd105*) | s | NM_007932 | TGA GCG TGT CTC CAT TGA CC | 416 | 11 | NIH Primer Tool |
| | as | | GGG GCC ACG TGT GTG AGA A | | 15 | |
| *Cd44* | s | NM_009851 | CAC CAT TTC CTG AGA CTT GCT | 148 | 18 | IDT Mm.PT.58.12084136 |
| | as | | TCT GAT TCT TGC CGT CTG C | | 19 | |
| *Pecam1* (*Cd31*) | s | NM_008816 | CTG CCA GTC CGA AAA TGG AAC | 218 | 7 | MGH Primer Bank ID 6679273a1 |
| | as | | CTT CAT CCA CTG GGG CTA TC | | 8 | |
| *Gjc1* (*Connexin 45*) | s | NM_008122 | GGT AAC AGG AGT TCT GGT GAA | 140 | 2 | IDT Mm.PT.58.8383900 |
| | as | | TCG AAA GAC AAT CAG CAC AGT | | 3 | |
| *Anoctamin 1* (*TMEM16A*) | s | NM_178642 | GGC ATT TGT CAT TGT CTT CCA G | 141 | 25 | IDT Real time primer tool |
| | as | | TCC TCA CGC ATA AAC AGC TC | | 26 | |
| *Ptprc* (*Cd45*) | s | NM_001111316 | ATG CAT CCA TCC TCG TCC AC | 225 | 29 | NIH Primer Tool |
| | as | | TGA CTT GTC CAT TCT GGG CG | | 31 | |

MGH Harvard Primer Bank (*Wang and Seed, 2003*; *Spandidos et al., 2008*; *Spandidos et al., 2010*).

## scRNAseq analysis of mouse IALVs

For scRNAseq analyses of isolated IALVs we used a total of 10 *Rosa26^mTmG^* mice, without Cre expression and without tamoxifen treatment, with equivalent representation of sex (five males and five females with ages between 10 and 12 months). Full-length IALVs from both the left and right sides of each *Rosa26^mTmG^* mouse were isolated and cleaned of excessive matrix and adipose tissue. Isolated vessels were digested into single-cell suspensions as described above, and the cells were kept on ice following single-cell suspension until all the tissues had been processed. Cells from all vessels were combined and sorted for tdTomato expression to remove debris and concentrate the cells for downstream single-cell 3′ RNA-Seq libraries creation with 10x Genomics Chromium Chip and Chromium Next GEM Single Cell 3′ RNA-Seq reagents. Samples were sequenced with the NovaSeq 6000 S4-PE100 flow cell.

*Mus musculus* genome GRCm39 and annotation GTF (v106) from Ensembl (https://useast.ensembl.org/Mus_musculus/Info/Index) were used to build the reference index and the reads were processed using Cell Ranger (v7.0.1; *Zheng et al., 2017*) with default parameters. The quality control and filtering steps were performed using R (v4.2.1; https://www.r-project.org/) as outlined in the Seurat pre-processing steps (*Hao et al., 2021*; *Hao et al., 2024*). Ambient RNA was removed from the Cell Ranger output with SoupX (*Young and Behjati, 2020*). Doublet score for each cell was estimated using scDBlFinder (v1.12.0; *Germain et al., 2021*). Non-expressed genes (sum zero across all samples) and low-quality cells (>10% mitochondrial genes, <500 genes, <1000 UMIs per cell and doublet score <0.5) were excluded from analysis. Of 10,188 cells, 7435 passed our inclusion criteria and included three dominant clusters including LECs (2962 cells), LMCs (978 cells), and fibroblasts (2261 cells) with the remaining cells comprising immune cells (1147 cells) and some mammary epithelial cell contamination (87 cells). Cells passing filtering were normalized/scaled (SCTransformation), dimensionally reduced (t-distributed stochastic neighbor embedding) and UMAP clustered, and hierarchically analyzed with Seurat (*Hao et al., 2021*; *Hao et al., 2024*) with default parameters. To select the optimal cluster resolution, we used Clustree with various resolutions. We examined the resulting tree to identify a resolution where the clusters were well separated and biologically meaningful, ensuring minimal merging or splitting at higher resolutions. Our goal was to find a resolution that captured relevant cell subpopulations while maintaining distinct clusters without excessive fragmentation. Initial clustering of the entire population of cells was done at a resolution of 0.8 and 18 PCs to achieve the UMAP of 0–19 cell clusters as shown in *Figure 5a*. We used a resolution of 0.5 for sub-clustering LMCs (original groups 5 and 6), 0.87 for LECs (original groups 0, 1, 2, and 11), and 1.0 for fibroblasts (3, 7, 8, 9, 10, and 13). Marker gene expression profile on cell clusters and gene co-expression was visualized using Seurat and ShinyCell R application (*Ouyang et al., 2021*). The full

scRNAseq raw dataset has been uploaded to the NIH GEO under the accession number GSE277843. Differential gene expression within subclusters of LECs, LMCs, and AdvCs was performed using Seurat's 'Find Markers' function and with a minimum of either 40% or 50% cell expression and average log fold change (Log2FC) minimum of 0.5 or 1. When assessing the LMC IP3 receptor genes *Itpr1-3* and *Itprid2*, a percent cell expression of 40% and Log2FC of 0.25 was used. In the volcano plot for LEC subcluster 8 differential gene expression, listed genes had a cutoff of a Log2FC of 2 or –2 to be displayed on the plot.

## Ex vivo Ca²⁺ imaging with the genetically encoded GCaMP6f indicator

CkitCreER$^{T2}$, Myh11CreER$^{T2}$, and PdgfrαCreER$^{TM}$ mice were crossed with GCaMP6f mice in a similar manner as described for *Rosa26*$^{mTmG}$. CkitCreER$^{T2}$-GCaMP6f, PdgfrαCreER$^{TM}$-GCaMP6f, and Myh11CreER$^{T2}$-GCaMP6f were induced with tamoxifen (10 mg/ml) for 5 consecutive days by i.p. injection. IALVs isolated from CkitCreER$^{T2}$-GCaMP6f, PdgfrαCreER$^{TM}$-GCaMP6f, and Myh11CreER$^{T2}$-GCaMP6f were cannulated as described above. The cannulated vessel, with micropipette holders, observation chamber, and V-track mounting system, was transferred to the stage of the spinning disk confocal with a Prime95B scMOS camera (Photometrics), a Cascade II EMCCD (Photometrics), or an Ixon888 EMCCD camera (Andor) for Ca²⁺ imaging (*Castorena-Gonzalez et al., 2018b*). Pressures for the input and output cannula were connected to a T-junction which was set briefly to 8 cmH$_2$O and the vessel lengthened to remove axial slack. A peristaltic pump maintained constant perfusion of the observation chamber with Krebs buffer at a rate of 0.5 ml/min while the vessel equilibrated at 37°C for 30–60 min with pressures set to 3 cmH$_2$O. Spontaneous contractions were allowed to stabilize over a period of 30 min to verify normal function and then were blunted with 2 μM wortmannin to limit movement associated with contractions during Ca²⁺ imaging. A Windows-based computer was used to digitize the pressure transducer signals and video image of the vessel from a firewire camera at 30–40 Hz (*Davis et al., 2012*). Real-time inner diameter tracking was made with LabVIEW (National Instruments; Austin, TX) (*Davis, 2005*). Once contractions were <5 μm in amplitude, Ca²⁺ recordings were made at 20 FPS for 20–40 s.

## Ca²⁺ imaging and analysis in IALVs over the contraction cycle

Background noise was determined by using the histogram feature of FIJI in a rectangle in a region of the field of view without sample. This value was subtracted from the entire field of view. In some cases, the vessel movement due to contraction was offset with video stabilization with the FIJI plugin Image Stabilizer. A max projection was used to create non-overlapping ROIs of GCaMP6f$^+$ cells for each inducible Cre-GCaMP6f IALV. From these cell ROIs, the 'reslice z' function was used to create pseudo-linescan STMs, which were divided by their baseline values to obtain $F/F_0$ values for each individual cell. At least three cells, except in the case of 1 CkitCreER$^{T2}$-GCaMP6f IALV, in which only two cells were observed, were analyzed in this manner for each vessel segment. Max projections of the image stack were then used to create non-overlapping cell masks of 3–5 muscle cells per field of view of one vessel. Ca²⁺ traces for those cells contained 5–10 contraction cycles and Ca²⁺ transients and were characterized for peak intensity (expressed as a baseline-referenced ratio, $F/F_0$), frequency, and duration in seconds.

## Analysis of subcellular Ca²⁺ transients in Myh11CreER$^{T2}$-GCaMP6f IALVs

For Myh11CreER$^{T2}$, we performed Ca²⁺ imaging as above in the presence of 1 μM nifedipine to stop the 'Ca²⁺ flashes' associated with APs (*Zawieja et al., 2018a*) and focus on the subcellular activity at three different experimental pressures of 0.5, 2, and 5 cmH$_2$O. For this protocol, we used a particle analysis approach to analyze all Ca²⁺ transients in the field of view. Ca²⁺ transients in intact vessels were quantified by particle analysis using Volumetry software (version G8d) as previously described (*Drumm et al., 2017*; *Drumm et al., 2019a*). Movies of Ca²⁺ transients in intact vessels were imported into Volumetry software (version G8d) and background subtracted. Movies were smoothed using a Gaussian filter: 1.5 × 1.5 mM, StdDev 1.0. Raw Ca²⁺ transients were converted to Ca²⁺ particles (PTCLs) using a flood-fill algorithm as previously described (*Drumm et al., 2017*; *Drumm et al., 2019a*). PTCLs <10 μM² were rejected to facilitate the removal of noise, and then the total PTCL area and PTCL count could be tabulated for each recording.

## Membrane potential recordings in IALVs

Mouse IALVs were isolated and cleaned as described above. IALVs were pressurized in our isobaric myography apparatus and allowed to equilibrate to ensure typical contractile activity was evident. A bolus of wortmannin at 2 µM was then applied to the bath to blunt contraction amplitude below 5 µm. Intracellular recordings of lymphatic muscle were made with microelectrodes (250–300 MΩ) filled with 1 M KCl and an SEC-05x amplifier (NPI) connected to a Grass S48 stimulator, viewed with a Tektronix TDS3052 digital oscilloscope. Membrane potential and diameter were simultaneously recorded using LabVIEW. Membrane potential and APS were allowed to stabilize and then pressure was slowly raised from 0.5 to 2 cmH$_2$O and then 5 cmH$_2$O. In some cases, the electrode dislodged due to the intrinsic contractions of the vessel or wall displacement as pressure was modulated. In these situations, we attempted to re-impale the cell or one of the neighboring cells. Only vessels in which a recording with a minimum of three stable APs was successfully recorded at two of the three experimental pressures were used for subsequent analysis.

We also confirmed LMC impalement using microelectrode filled with 1 M KCl and (100 µg/ml) Biocytin-AF488 (A12924, Thermo Fisher) to label impaled cells that displayed APs, over a 10-min recording period. Following the impalement and loading with Biocytin-AF488, the vessel was transferred to our imaging apparatus for confocal imaging and 3D reconstruction using the Andor Dragonfly 200 and IMARIS. Image stacks were taken with a 25x water objective at 0.5 micron intervals throughout the diameter of the vessel.

## Light activation of ChR2 to stimulate popliteal cLV contractions

As the IALV has a nearly continuous contractile cycle, we utilized the popliteal vessel for its much slower contraction frequency in the experiments testing our ability to trigger a propagated contraction upon stimulation of the enforced expression of ChR2. Popliteal vessels were isolated from Ckit-CreER$^{T2}$-ChR2/tdTomato, PdgfrαCreER$^{TM}$-ChR2/tdTomato, or Myh11CreER$^{T2}$-ChR2/tdTomato mice (3–9 months of age) as previously described (*Scallan and Davis, 2013*), although we intentionally retained some connective tissue and adipose tissue to ensure we had a sufficient population of recombined cells to test in the adventitia layer of the vessel. Contractions were allowed to stabilize over a 30-min equilibration period with pressure set to 3 cmH$_2$O. If basal contraction frequency was too high, we applied pinacidil to the bath in 100 nM increments, without exceeding 600 nM, to further slow contraction frequency to around 6 contractions per minute. Pinacidil at sub 1 µM doses can slow contraction frequency without causing overt hyperpolarization of membrane potential (*Davis et al., 2020*). Supplemental 100 nM doses of pinacidil were applied throughout the experiment to maintain a spontaneous contraction frequency below 6 per minute to allow ample diastolic time for ChR2 stimulation. Throughout this protocol, the popliteal was allowed to contract spontaneously to ensure we had not overly inhibited APs by the pacemaking cells with pinacidil. Occasionally, spontaneous contractions occurred just prior to light-evoked contractions, resulting in a potential false positive, so we performed multiple stimulations over a period of 5–10 min, typically waiting at least 3 s after any spontaneous contraction before stimulating. Care was made to align the light fiber in such a way that only part of the vessel would be directly illuminated and so target cells of interest would be directly activated by 473 nm light using a Laser diode (Doric LD Fiber Light Source, Quebec, Canada), through an optical probe with a 10-µm tip (Doric, OPT_200_0.22.010). To further limit the excitation field, the optical probe was coated with black acrylic paint using an eyelash brush so that the uncoated opening was ~2–3 µm. With the probe positioned within 5 µm of one side of the vessel wall, the spread of light covered an area ~10–100 µm wide on the back side of the vessel (depending on the diode amplitude setting). Light pulses, 200 ms in length, were triggered by a Grass S9 stimulator (Harvard Apparatus, Holliston, MA) connected to the external TTL input of the laser diode. Pulse amplitude was adjusted between 40 and 90 mA using the Laser Diode Module Driver (Doric). A contraction was considered to be triggered if it occurred within 50ms of stimulation. We performed photo-stimulation from 2 to 4 sites within each vessel, with 6–14 stimulations per site. If a photo-stimulation was triggered incidentally after the initiation of a 'spontaneous contraction', it was discarded from the analysis. For Myh11CreER$^{T2}$-ChR2-tdTomato, six vessels from three separate mice were tested. For PdgfrαCreER$^{TM}$-ChR2-tdTomato, six vessels from four separate mice were tested with a max of two vessels per mouse. For CkitCreER$^{T2}$-ChR2-tdTomato, seven vessels from three separate mice were assessed. Diameter was recorded to align photo-activation with the contraction cycle using LabVIEW.

## Solutions and chemicals

Krebs buffer was composed of (in mM): 146.9 NaCl, 4.7 KCl, 2 CaCl$_2$, 1.2 MgSO$_4$, 1.2 NaH$_2$PO$_4$·H$_2$O, 3 NaHCO$_3$, 1.5 NaHEPES, and 5 D-glucose (pH = 7.4 at 37°C). Krebs-BSA buffer was prepared with the addition of 0.5% (wt/vol) BSA while Krebs Ca$^{2+}$-free replaced CaCl$_2$ with 3 mM EGTA. Tamoxifen was dissolved to 10 mg/ml in a Safflower Oil-Ethanol (95–5% vol/vol) solution with rocking agitation, separated into aliquots, and stored at –20°C. Wortmannin was dissolved in DMSO to a stock solution of 1 mM. Pinacidil was dissolved in DMSO to a stock concentration of 1 μM. Nifedipine was dissolved in DMSO to a stock concentration of 1 mM. All chemicals were obtained from Sigma (St. Louis, MO), except for BSA (US Biochemicals; Cleveland, OH), MgSO$_4$, and NaHEPES (Fisher Scientific; Pittsburgh, PA).

## Statistical tests

Statistical differences in the isobaric contractile tests for popliteal cLVs isolated from PdgfrαCreER$^{TM}$-Ano1$^{fl/fl}$, PdgfrαCreER$^{TM}$-Gjc1$^{fl/fl}$, and PdgfrαCreER$^{TM}$-Cacna1c$^{fl/fl}$ mice over the various contractile parameters were assessed via (1) repeated measures two-way ANOVAs with Sidak's multiple comparison tests performed using Prism9 (Graphpad). Data are plotted as mean ± SEM and significance determined at $p < 0.05$ and $0.10 > p > 0.05$ were reported. Cre negative mice were used for controls, and experiment order was not randomized aside from random mouse selection from cages housing both Cre$^+$ or Cre$^-$ mice. Experimental sample size was determined by the results from our previous experiments assessing Gjc1, Ano1, Cacna1c with Myh11CreER$^{T2}$ mice. Data from cLVs in which a negative tone value was recorded at any pressure, which typically indicated incomplete relaxation or occluding bubbles in the cannula, were not included in the tone analysis. Vessels that failed to contract at a given pressure had no value recorded for ejection fraction or normalized amplitude, and REML mixed effects model was used in place of repeated measures two-way ANOVA. We used a categorical Chi-squared statistical test for the experiments assessing our ability to trigger a contraction with activation of ChR2+ cells. Ca$^{2+}$ particle area and frequency were compared using one-way ANOVA with Tukey's post hoc test. Significance was determined at $p < 0.05$. A mixed effects analysis with Tukey's multiple comparison post hoc test was used to compare AP parameters across pressure using Prism9 (GraphPad).

## Acknowledgements

We would like to thank Stefan Offermanns for the donation of the Myh11CreER$^{T2}$ mice, Klaus Willecke for his donation of Gjc1$^{f/f}$ mice, Dieter Sauer for his donation of CkitCreER$^{T2}$ mice, Ralph Adams for his donation of the PdgfrβCreER$^{T2}$ mice, and Young Hong (University of Southern California) for his donation of the Prox1-eGFP mice (Choi et al., 2011).

This work was supported by NIH HL-122608 and HL-122578 to MJD, HL-143198 and HL-175083 SDZ, and HL-141143 and HL-168568 to JAC-G, and AHA CDA-931652 to CEN. A Patro was supported by MizzouForward Undergraduate Research Fellow.

## Additional information

### Funding

| Funder | Grant reference number | Author |
| --- | --- | --- |
| National Heart, Lung, and Blood Institute | HL-175083 | Scott D Zawieja |
| National Heart, Lung, and Blood Institute | HL-143198 | Scott D Zawieja |
| National Heart, Lung, and Blood Institute | HL-122608 | Michael J Davis |
| National Heart, Lung, and Blood Institute | HL-122578 | Michael J Davis |

| Funder | Grant reference number | Author |
|---|---|---|
| National Heart, Lung, and Blood Institute | HL-141143 | Jorge A Castorena-Gonzalez |
| National Heart, Lung, and Blood Institute | HL-168568 | Jorge A Castorena-Gonzalez |
| American Heart Association | CDA-931652 | Charles E Norton |

The funders had no role in study design, data collection, and interpretation, or the decision to submit the work for publication.

## Author contributions

Scott D Zawieja, Conceptualization, Resources, Data curation, Formal analysis, Supervision, Funding acquisition, Investigation, Visualization, Methodology, Writing – original draft, Project administration, Writing – review and editing; Grace A Pea, Sarah E Broyhill, Advaya Patro, Karen H Bromert, Hae Jin Kim, Min Li, Data curation, Formal analysis, Investigation, Writing – review and editing; Charles E Norton, Investigation, Writing – review and editing; Sathesh Kumar Sivasankaran, Data curation, Formal analysis, Investigation, Methodology, Writing – review and editing; Jorge A Castorena-Gonzalez, Data curation, Formal analysis, Funding acquisition, Validation, Investigation, Methodology, Writing – review and editing; Bernard T Drumm, Data curation, Software, Formal analysis, Investigation, Visualization, Methodology, Writing – review and editing; Michael J Davis, Conceptualization, Resources, Data curation, Software, Formal analysis, Supervision, Funding acquisition, Investigation, Visualization, Methodology, Writing – original draft, Project administration

## Author ORCIDs

Scott D Zawieja (iD) https://orcid.org/0000-0003-4658-3179

## Ethics

All animal protocols were approved by the University of Missouri Animal Care and Use Committee (Protocol 53461 and 41500) and conformed to the US Public Health Service policy for the humane care and use of laboratory animals (PHS Policy, 1996).

Reviewer #1 (Public review): https://doi.org/10.7554/eLife.90679.4.sa1
Reviewer #2 (Public review): https://doi.org/10.7554/eLife.90679.4.sa2
Reviewer #3 (Public review): https://doi.org/10.7554/eLife.90679.4.sa3
Author response https://doi.org/10.7554/eLife.90679.4.sa4

# Additional files

## Supplementary files

MDAR checklist

## Data availability

All data generated or analyzed during this study are included in the manuscript and supporting files. Source data files have been provided for all gel electrophoresis results. The scRNA-seq dataset has been uploaded to the NIH GEO #GSE277843. The authors declare that all other data supporting the findings of this study are available within the paper, its supplementary information files, and the uploaded scRNA-seq dataset.

The following dataset was generated:

| Author(s) | Year | Dataset title | Dataset URL | Database and Identifier |
|---|---|---|---|---|
| Zawieja SD, Pea GA, Broyhill SE, Patro A, Bromert KH, Norton CE, Kim HJ, Sivasankaran SK, Li M, Castorena-Gonzalez JA, Drumm BT, Davis MJ | 2024 | Characterization of the cellular components of mouse collecting lymphatic vessels reveals that lymphatic muscle cells are the innate pacemaker cells regulating lymphatic contractions | https://www.ncbi.nlm.nih.gov/geo/query/acc.cgi?acc=GSE277843 | NCBI Gene Expression Omnibus, GSE277843 |

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

# Appendix 1

## Appendix 1—key resources table

| Reagent type (species) or resource | Designation | Source or reference | Identifiers | Additional information |
|---|---|---|---|---|
| Strain, strain background (*Mus musculus*) | C57BL/6J | The Jackson Laboratory | Jax Strain #000664<br>RRID:IMSR_JAX:000664 | |
| Strain, strain background (*Mus musculus*) | Rosa26$^{mTmG}$ | The Jackson Laboratory | Jax Strain #007676<br>B6.129(Cg)-Gt(ROSA)26Sortm4(ACTB-tdTomato,-EGFP) Luo/J<br>RRID:IMSR_JAX:007676 | |
| Strain, strain background (*Mus musculus*) | PdgfrαCre | The Jackson Laboratory | Jax Strain #013148<br>C57BL/6-Tg(Pdgfra-cre)1Clc/J<br>RRID:IMSR_JAX: 013148 | |
| Strain, strain background (*Mus musculus*) | Csfr1-EGFP | The Jackson Laboratory | Jax Strain #018549<br>B6.Cg-Tg(Csf1r-EGFP)1Hume/J<br>RRID:IMSR_JAX:018549 | |
| Strain, strain background (*Mus musculus*) | PdgfrαCreER$^{TM}$ | The Jackson Laboratory | Jax Strain #018280<br>B6N.Cg-Tg(Pdgfra-cre/ERT)467Dbe/J<br>RRID:IMSR_JAX:018280 | |
| Strain, strain background (*Mus musculus*) | Cspg4-Cre | The Jackson Laboratory | Jax Strain #:008533<br>B6;FVB-Ifi208Tg(Cspg4-cre)1Akik/J<br>RRID:IMSR_JAX:008533 | |
| Strain, strain background (*Mus musculus*) | ChR2/tdTomato | The Jackson Laboratory | Jax Strain #012567<br>B6.Cg-Gt(ROSA)26Sortm27.1(CAG-COP4*H134R/tdTomato)Hze/J<br>RRID:IMSR_JAX:012567 | |
| Strain, strain background (*Mus musculus*) | PdgfrβCreER$^{T2}$ | The Jackson Laboratory | Jax Strain #029684<br>B6.Cg-Tg(Pdgfrb-cre/ERT2)6096Rha/J<br>RRID:IMSR_JAX:029684 | |
| Strain, strain background (*Mus musculus*) | Myh11CreER$^{T2}$ | The Jackson Laboratory | Jax Strain #019079<br>B6.FVB-Tg(Myh11-icre/ERT2)1Soff/J<br>RRID:IMSR_JAX:019079 | |
| Strain, strain background (*Mus musculus*) | CkitCreER$^{T2}$ | Dieter Saur (Technical University of Munich) | Kittm1(cre/ERT2)Dsa | |
| Strain, strain background (*Mus musculus*) | Prox1-eGFP | Young-Kwon Hong (University of Southern California) | MMRRC ID #31006<br>Tg(Prox1-EGFP)KY221Gsat/Mmucd<br>RRID:MMRRC_031006-UCD | |
| Strain, strain background (*Mus musculus*) | GCaMP6f | Jax | Jax Strain #028865<br>Ai95(RCL-GCaMP6f)-D (C57BL/6J) or Ai95D (C57BL/6J)<br>RRID:IMSR_JAX:028865 | |
| Antibody | anti-ACTA2 | Sigma-Aldrich | Cat# A2547, RRID:AB_476701 | (IF) 1:500 |
| Antibody | anti-GFP | Thermo Fisher Scientific | Cat# A-11122, RRID:AB_221569 | (IF) 1:200 |
| Antibody | anti-CKIT | Cell Signaling | Cell Signaling Technology Cat# 3074, RRID:AB_1147633 | (IF) 1:100 |
| Antibody | anti-VIMENTIN | Thermo Fisher Scientific | Cat# OMA1-06001, RRID:AB_325529 | (IF) 1:100 |
| Antibody | anti-DESMIN | Invitrogen | Cat# PA5-16705, RRID:AB_10977258 | (IF) 1:200 |
| Antibody | anti-GFP | Abcam | Cat# ab13970, RRID:AB_300798 | (IF) 1:200 |
| Antibody | anti-CD34 (RAM34) | Thermo Fisher Scientific | Cat# 14-0341-82, RRID:AB_467210 | (IF) 1:200 |
| Antibody | anti-PDGFRα | R&D Systems | Cat# AF1062, RRID:AB_2236897 | (IF) 1:200 |
| Antibody | anti-PDGFRβ | eBiosciences | Cat# 14-1402-82, RRID:AB_467493 | (IF) 1:200 |
| Antibody | anti-CALPONIN | Abcam | Cat# ab46794, RRID:AB_2291941 | (IF) 1:500 |
| Antibody | anti-LY6A | Biolegend | Cat# 108101, RRID:AB_313338 | (IF) 1:200 |

*Appendix 1 Continued on next page*

*Appendix 1 Continued*

| Reagent type (species) or resource | Designation | Source or reference | Identifiers | Additional information |
|---|---|---|---|---|
| Sequence-based reagent | *Prox1-*Forward NM_008937 | NIH Primer Tool, this paper | PCR primers | GTA AGA CAT CAC CGC GTG C |
| Sequence-based reagent | *Prox1-*Reverse NM_008937 | NIH Primer Tool, this paper | PCR primers | TCA TGG TCA GGC ATC ACT GG |
| Sequence-based reagent | *Itgam* Reverse NM_008401 | MGH Primer Bank ID 668048a1 | PCR primers | ATG GAC GCT GAT GGC AAT ACC |
| Sequence-based reagent | *Itgam* Forward NM_008401 | MGH Primer Bank ID 668048a1 | PCR primers | TCC CCA TTC ACG TCT CCC A |
| Sequence-based reagent | *Pdgfra* Forward NM_011058 | MGH Primer Bank ID 26349287a1 | PCR primers | AGA GTT ACA CGT TTG AGC TGT C |
| Sequence-based reagent | *Pdgfra* Reverse NM_011058 | MGH Primer Bank ID 26349287a1 | PCR primers | GTC CCT CCA CGG TAC TCC T |
| Sequence-based reagent | *Myh11* Forward NM_013607 | MGH Primer Bank ID 7305295a1 | PCR primers | AAG CTG CGG CTA GAG GTC A |
| Sequence-based reagent | *Myh11* Reverse NM_013607 | MGH Primer Bank ID 7305295a1 | PCR primers | CCC TCC CTT TGA TGG CTG AG |
| Sequence-based reagent | *Ckit* Forward NM_021099 | *Drumm et al., 2018* | PCR primers | CGC CTG CCG AAA TGT ATG ACG |
| Sequence-based reagent | *Ckit* Reverse NM_021099 | *Drumm et al., 2018* | PCR primers | GGT TCT CTG GGT TGG GGT TGC |
| Sequence-based reagent | *Pdgfrb* Forward NM_008809 | *Basciani et al., 2004* | PCR primers | AGC TAC ATG GCC CCT TAT GA |
| Sequence-based reagent | *Pdgfrb* Reverse NM_008809 | *Basciani et al., 2004* | PCR primers | GGA TCC CAA AAG ACC AGA CA |
| Sequence-based reagent | *Cdh5* Forward NM_009868 | IDT Primer Quest Tool, this paper | PCR primers | CTT CCT TAC TGC CCT CAT TGT |
| Sequence-based reagent | *Cdh5* Reverse NM_009868 | IDT Real time primer tool, this paper | PCR primers | CTG TTT CTC TCG GTC CAA GTT |
| Sequence-based reagent | *Nos3* Forward NM_008713 | IDT Real time primer tool, this paper | PCR primers | CTG CCA CCT GAT CCT AAC TTG |
| Sequence-based reagent | *Nos3* Reverse NM_008713 | IDT Real time primer tool, this paper | PCR primers | CAG CCA AAC ACC AAA GTC ATG |
| Sequence-based reagent | *Acta2* (Smooth Muscle Actin) Forward NM_007392 | IDT TaqMan Mm.PT.58.16320644 | PCR primers | GAG CTA CGA ACT GCC TGA C |
| Sequence-based reagent | *Acta2* (Smooth Muscle Actin) Reverse NM_007392 | IDT TaqMan Mm.PT.58.16320644 | PCR primers | CTG TTA TAG GTG GTT TCG TGG A |

*Appendix 1 Continued*

| Reagent type (species) or resource | Designation | Source or reference | Identifiers | Additional information |
|---|---|---|---|---|
| Sequence-based reagent | *Cacna1c* (CaV 1.2) Forward NM_001159533 | *Cheng et al., 2007* | PCR primers | ATG GTC AAT GAA AAC ACG AGG ATG |
| Sequence-based reagent | *Cacna1c* (CaV 1.2) Reverse NM_001159533 | *Cheng et al., 2007* | PCR primers | GGA ACT GAC GGT AGA GAT GGT TGC |
| Sequence-based reagent | *Cd34* Forward NM_133654 | IDT Mm.PT.58.8626728 | PCR primers | GGT ACA GGA GAA TGC AGG TC |
| Sequence-based reagent | *Cd34* Reverse NM_133654 | IDT Mm.PT.58.8626728 | PCR primers | CGT GGT AGC AGA AGT CAA GT |
| Sequence-based reagent | *Cspg4* (Ng2) Forward NM_139001 | IDT Mm.PT.58.29461721 | PCR primers | CTT CAC GAT CAC CAT CCT TCC |
| Sequence-based reagent | *Cspg4* (Ng2) Reverse NM_139001 | IDT Mm.PT.58.29461721 | PCR primers | CCC GAA TCA TTG TCT GTT CCC |
| Sequence-based reagent | *Vimentin* Forward NM_011701 | *Li et al., 2016* | PCR primers | CTG TAC GAG GAG GAG ATG CG |
| sequence-based reagent | *Vimentin* Reverse NM_011701 | *Li et al., 2016* | PCR primers | AAT TTC TTC CTG CAA GGA TT |
| Sequence-based reagent | *Desmin* Forward NM_010043 | MGH Primer Bank ID 33563250a1 | PCR primers | GTG GAT GCA GCC ACT CTA GC |
| Sequence-based reagent | *Desmin* Reverse NM_010043 | MGH Primer Bank ID 33563250a1 | PCR primers | TTA GCC GCG ATG GTC TCA TAC |
| Sequence-based reagent | *Mcam* Forward NM_023061 | MGH Primer Bank ID 10566955a1 | PCR primers | CCC AAA CTG GTG TGC GTC TT |
| Sequence-based reagent | *Mcam* Reverse NM_023061 | MGH Primer Bank ID 10566955a1 | PCR primers | GGA AAA TCA GTA TCT GCC TCT CC |
| Sequence-based reagent | *Klf4* Forward NM_010637 | *Majesky et al., 2017* | PCR primers | ATT AAT GAG GCA GCC ACC TG |
| Sequence-based reagent | *Klf4* Reverse NM_010637 | *Majesky et al., 2017* | PCR primers | GGA AGA CGA GGA TGA AGC TG |
| Sequence-based reagent | *Ly6a* Forward NM_001271416 | *Majesky et al., 2017* | PCR primers | CTC TGA GGA TGG ACA CTT CT |
| Sequence-based reagent | *Ly6a* Reverse NM_001271416 | *Majesky et al., 2017* | PCR primers | GGT CTG CAG GAG GAC TGA GC |
| Sequence-based reagent | *Gli1* Forward NM_01029 | *Kramann et al., 2015* | PCR primers | ATC ACC TGT TGG GGA TGC TGG AT |
| Sequence-based reagent | *Gli1* Reverse NM_01029 | *Kramann et al., 2015* | PCR primers | CGT GAA TAG GAC TTC CGA CAG |

*Appendix 1 Continued on next page*

*Appendix 1 Continued*

| Reagent type (species) or resource | Designation | Source or reference | Identifiers | Additional information |
|---|---|---|---|---|
| Sequence-based reagent | *Itgb1* Forward NM_010578 | NIH Primer Tool, this paper | PCR primers | TCG ATC CTG TGA CCC ATT GC |
| Sequence-based reagent | *Itgb1* Reverse NM_010578 | NIH Primer Tool, this paper | PCR primers | AAC AAT TCC AGC AAC CAC GC |
| Sequence-based reagent | *Eng* Forward NM_007932 | NIH Primer Tool, this paper | PCR primers | TGA GCG TGT CTC CAT TGA CC |
| Sequence-based reagent | *Eng* Reverse NM_007932 | NIH Primer Tool, this paper | PCR primers | GGG GCC ACG TGT GTG AGA A |
| Sequence-based reagent | *Cd44* Forward NM_009851 | IDT Mm.PT.58.12084136 | PCR primers | CAC CAT TTC CTG AGA CTT GCT |
| Sequence-based reagent | *Cd44* Reverse NM_009851 | IDT Mm.PT.58.12084136 | PCR primers | TCT GAT TCT TGC CGT CTG C |
| Sequence-based reagent | *Pecam1* Forward NM_008816 | MGH Primer Bank ID 6679273a1 | PCR primers | CTG CCA GTC CGA AAA TGG AAC |
| Sequence-based reagent | *Pecam1* Reverse NM_008816 | MGH Primer Bank ID 6679273a1 | PCR primers | CTT CAT CCA CTG GGG CTA TC |
| Sequence-based reagent | *Gjc1* Forward NM_008122 | IDT Mm.PT.58.8383900 | PCR primers | GGT AAC AGG AGT TCT GGT GAA |
| Sequence-based reagent | *Gjc1* Reverse NM_008122 | IDT Mm.PT.58.8383900 | PCR primers | TCG AAA GAC AAT CAG CAC AGT |
| Sequence-based reagent | *Anoctamin 1* Forward NM_178642 | IDT Real time primer tool, this paper | PCR primers | GGC ATT TGT CAT TGT CTT CCA G |
| Sequence-based reagent | *Anoctamin 1* Reverse NM_178642 | IDT Real time primer tool, this paper | PCR primers | TCC TCA CGC ATA AAC AGC TC |
| Sequence-based reagent | *Ptprc* Forward NM_001111316 | NIH Primer Tool, this paper | PCR primers | ATG CAT CCA TCC TCG TCC AC |
| Sequence-based reagent | *Ptprc* Reverse NM_001111316 | NIH Primer Tool, this paper | PCR primers | TGA CTT GTC CAT TCT GGG CG |
| Chemical compound, drug | Methylene Blue | Sigma | M9140 | |
| Chemical compound, drug | Blockaid | Thermo Fisher | A11122 | |
| Peptide, recombinant protein | Collagenase H | Sigma | C8051 | |
| Peptide, recombinant protein | Collagenase F | Sigma | C7926 | |
| Chemical compound, drug | Dithioerythritol (DTT) | Sigma | D8161 | |
| Peptide, recombinant protein | Elastase | Worthington | LS00635 | |
| Peptide, recombinant protein | Papain | Sigma | P4762 | |
| Antibody | Donkey anti-mouse AF647 | Thermo Fisher | A32787 | (IF) 1:200 |

*Appendix 1 Continued on next page*

*Appendix 1 Continued*

| Reagent type (species) or resource | Designation | Source or reference | Identifiers | Additional information |
|---|---|---|---|---|
| Antibody | Donkey anti-Rat AF555 | Thermo Fisher | A48270 | (IF) 1:200 |
| Antibody | Donkey anti-Rabbit AF488 | Thermo Fisher | A21206 | (IF) 1:200 |
| Antibody | Donkey anti-Goat AF647 | Thermo Fisher | A21447 | (IF) 1:200 |
| Antibody | Donkey anti-Goat AF555 | Thermo Fisher | A21432 | (IF) 1:200 |
| Commercial assay or kit | Arcturus PicoPure RNA isolation kit | Thermo Fisher | KIT0204 | |
| Software, algorithm | Prism 10 | GraphPad | | |
| Software, algorithm | Volumetry software (version G8d) | Grant Hennig | | *Drumm et al., 2017; Drumm et al., 2019a* |
| Software, algorithm | Seurat v4,v5 | | RRID:SCR_016341 | *Hao et al., 2021; Hao et al., 2024* |
| Software, algorithm | FIJI BIOP-JACoP | | | *Bolte and Cordelières, 2006* |

