## [Editor Report · eLife Assessment]

This manuscript aims to identify the pacemaker cells in the lymphatic collecting vessels - the cells that initiate the autonomous action potentials and contractions needed to drive lymphatic pumping. Through the exemplary use of existing approaches (genetic deletions and cytosolic calcium detection in multiple cell types), the authors **convincingly** determine that lymphatic muscle cells are the origin of the action potential that triggers lymphatic contraction. The inclusion of scRNAseq and membrane potential data enhances a tremendous study. This **fundamental** discovery establishes a new standard for the field of lymphatic physiology.

---

## [Referee Report · Reviewer #1 (Public review)]

Summary:

This manuscript explores the multiple cell types present in the wall of murine collecting lymphatic vessels with the goal of identifying cells that initiate the autonomous action potentials and contractions needed to drive lymphatic pumping. Through the use of genetic models to delete individual genes or detect cytosolic calcium in specific cell types, the authors convincingly determine that lymphatic muscle cells are the origin of the action potential that triggers lymphatic contraction.

Strengths:

The experiments are rigorously performed, the data justify the conclusions and the limitations of the study are appropriately discussed.

There is a need to identify therapeutic targets to improve lymphatic contraction and this work helps identify lymphatic muscle cells as potential cellular targets for intervention.

Comments on revisions: The authors have addressed all of the reviewer comments. They should be congratulated on their precise and comprehensive study.

---

## [Referee Report · Reviewer #2 (Public review)]

Summary:

This is a well written manuscript describing studies directed at identifying the cell type responsible for pacemaking in murine collecting lymphatics. Using state of the art approaches, the authors identified a number of different cell types in the wall of these lymphatics and then using targeted expression of Channel Rhodopsin and GCaMP, the authors convincingly demonstrate that only activation of lymphatic muscle cells produces coordinated lymphatic contraction and that only lymphatic muscle cells display pressure-dependent Ca2+ transients as would be expected of a pacemaker in these lymphatics.

Strengths:

The use of targeted expression of channel rhodopsin and GCaMP to test the hypothesis that lymphatic muscle cells serve as the pacemakers in musing lymphatic collecting vessels.

Weaknesses:

The only significant weakness was the lack of quantitative analysis of most of the imaging data shown in Figures 1-11. In particular the colonization analysis should be extended to show cells not expected to demonstrate colocalization as a negative control for the colocalization analysis that the authors present. These weaknesses have been resolved by revision and addition of new and novel RNAseq data, additional colocalization data and membrane potential measurements.

Comments on revisions: No additional concerns.

---

## [Referee Report · Reviewer #3 (Public review)]

Summary:

Zawieja et al. aimed to identify the pacemaker cells in the lymphatic collecting vessels. Authors have used various Cre-based expression systems and optogentic tools to identify these cells. Their findings suggest these cells are lymphatic muscle cells that drive the pacemaker activity in the lymphatic collecting vessels.

Strengths:

The authors have used multiple approaches to test their hypothesis. Some findings are presented as qualitative images, while some quantitative measurements are provided.

Weaknesses:

- More quantitative measurements.

- Possible mechanisms associated with the pacemaker activity.

- Membrane potential measurements.

Comments on revisions: I do not have any additional comments.

---

## [Author Response]

The following is the authors’ response to the previous reviews

**Reviewer #1 (Recommendations for the authors):**
The authors have done an impressive job in responding to the previous critique and even gone beyond what was asked. I have only very minor comments on this excellent manuscript. The manuscript also needs some light editing for grammar and readability.

We have worked to improve the grammar and readability of the manuscript.

Comments:Lines 227-234: At what age was tamoxifen administered to the various CreERTM mice?

We have updated the ages of the mice used in this study in the methods sections.

UMAP in Figure 5A is missing label for cluster 19.

The UMAP in Figure 5A has the label for cluster 19 at the center-bottom of the image.

Supplement Figure 6: Cluster 10 seems to be separate from the other AdvC clusters, and it includes some expression of Myh11 and Notch3. Further, there is low expression of Pdgfra in this cluster, which can be seen in panel B and panels D-I. Are the Pdgfra negative cells in the pie charts from cluster 10? Could the cells in this cluster by more LMC like than AdvC like?

We agree with the reviewer that the subcluster 10 of the fibroblasts cells are intriguing if only a minor population. When assessing just this population of cells, which is 77 cells out of 2261 total, 40 of the 77 were Pdgfra+ and of the 37 remaining Pdgfra- but 11 of those were still CD34+. Thus at least half of these cells could be expected to have the PdgfraCreERTM. Only 8 of the 37 were Pdgfra-Notch3+ while 12 cells were Pdgfra+Notch3+, and only 3 were Pdgfra-Myh11+ while 3 were Pdgfra+Myh11+. 26 of 77 cells were Pdgfra+Pdgfrb+ double positive, while 12 of 37 Pdgfra- cells were still Pdgfrb+. Additionally, within the 77 cells of subcluster 10 17 were positive for Scn3a (Nav1.3), 21were positive for Kcnj8 (Kir6.1), and 33 were positive for Cacna1c (Cacna1c) which are typically LMC markers would support the reviewers thinking that this group contains a fibroblast-LMC transitional cell type. Only 2 of 77 cells were positive for the BK subunit (Kcnma1), which is a classic smooth muscle marker. Another possibility is this population represents the Pdgfra+Pdgfrb+ valve interstitial cells we identified in our IF staining and in our reporter mice. Of note almost all cells in this cluster were Col3a1+ and Vim+. Even though we performed QC analysis to remove doublets, it is also possible some of these cells could represent doublets or contaminants, however the low % of Myh11 expression, a very highly expressed gene in LMCs especially compared to ion channels, would suggest this is less likely. Assessing the presence of this particular cell cluster in future RNAseq or with spatial transcriptomics will be enlightening.

Line 360. Proofread section title.

We have simplified this title to read “Optogenetic Stimulation of iCre-driven Channel Rhodopsin 2”

Lines 370-371. Are the length units supposed to be microns or millimeters?

We have corrected this to microns as was intended. Thank you for catching this error.

The resolution for each UMAP analysis should be stated, particularly for the identification of subclusters. How was the resolution chosen?

To select the optimal cluster resolution, we used Clustree with various resolutions. We examined the resulting tree to identify a resolution where the clusters were well-separated and biologically meaningful, ensuring minimal merging or splitting at higher resolutions. Our goal was to find a resolution that captures relevant cell subpopulations while maintaining distinct clusters without excessive fragmentation. We have now stated the resolution for the subclustering of the LECs, LMCs, and fibroblasts. We have also added greater detail regarding the total number of cells, QC analysis, and the marker identification criteria used to the methods sections. We used resolution of 0.5 for sub-clustering LMCs, 0.87 for LECs, and 1.0 for fibroblasts. These details are now added to the manuscript.